# *Stxbp1/Munc18-1* haploinsufficiency impairs inhibition and mediates key neurological features of *STXBP1* encephalopathy

Wu Chen[1,2], Zhao-Lin Cai[1,2], Eugene S Chao[1,2], Hongmei Chen[1,2], Colleen M Longley[2,3], Shuang Hao[4,5], Hsiao-Tuan Chao[1,4,5,6,7], Joo Hyun Kim[1,2], Jessica E Messier[1,2], Huda Y Zoghbi[1,3,4,5,6,8], Jianrong Tang[4,5], John W Swann[1,2,4], Mingshan Xue[1,2,3,6]*

[1]Department of Neuroscience, Baylor College of Medicine, Houston, United States; [2]The Cain Foundation Laboratories, Jan and Dan Duncan Neurological Research Institute at Texas Children's Hospital, Houston, United States; [3]Program in Developmental Biology, Baylor College of Medicine, Houston, United States; [4]Department of Pediatrics, Division of Neurology and Developmental Neuroscience, Baylor College of Medicine, Houston, United States; [5]Jan and Dan Duncan Neurological Research Institute at Texas Children's Hospital, Houston, United States; [6]Department of Molecular and Human Genetics, Baylor College of Medicine, Houston, United States; [7]McNair Medical Institute, The Robert and Janice McNair Foundation, Houston, United States; [8]Howard Hughes Medical Institute, Baylor College of Medicine, Houston, United States

*For correspondence:
Mingshan.Xue@bcm.edu

**Abstract** Mutations in genes encoding synaptic proteins cause many neurodevelopmental disorders, with the majority affecting postsynaptic apparatuses and much fewer in presynaptic proteins. Syntaxin-binding protein 1 (STXBP1, also known as MUNC18-1) is an essential component of the presynaptic neurotransmitter release machinery. De novo heterozygous pathogenic variants in *STXBP1* are among the most frequent causes of neurodevelopmental disorders including intellectual disabilities and epilepsies. These disorders, collectively referred to as *STXBP1* encephalopathy, encompass a broad spectrum of neurologic and psychiatric features, but the pathogenesis remains elusive. Here we modeled *STXBP1* encephalopathy in mice and found that *Stxbp1* haploinsufficiency caused cognitive, psychiatric, and motor dysfunctions, as well as cortical hyperexcitability and seizures. Furthermore, *Stxbp1* haploinsufficiency reduced cortical inhibitory neurotransmission via distinct mechanisms from parvalbumin-expressing and somatostatin-expressing interneurons. These results demonstrate that *Stxbp1* haploinsufficient mice recapitulate cardinal features of *STXBP1* encephalopathy and indicate that GABAergic synaptic dysfunction is likely a crucial contributor to disease pathogenesis.

## Introduction

Human genetic studies of neurodevelopmental disorders continue to uncover pathogenic variants in genes encoding synaptic proteins (*Hoischen et al., 2014*; *Zhu et al., 2014*; *Deciphering Developmental Disorders Study, 2015*; *Deciphering Developmental Disorders Study, 2017*; *Stessman et al., 2017*; *Lindy et al., 2018*), demonstrating the importance of these proteins for neurologic and psychiatric features. The molecular and cellular functions of many of

these synaptic proteins have been extensively studied. However, to understand the pathological mechanisms underlying these synaptic disorders, in-depth neurological and behavioral studies in animal models are necessary. While it is difficult to perform such studies for all disorders, this knowledge gap can be significantly narrowed by studying a few prioritized genes that are highly penetrant and affect a broad spectrum of neurologic and psychiatric features common among neurodevelopmental disorders (*Hoischen et al., 2014*; *Ogden et al., 2016*). Syntaxin-binding protein 1 (STXBP1, also known as MUNC18-1) is one such example because its molecular and cellular functions are well understood (*Rizo and Xu, 2015*), its pathogenic variants are emerging as prevalent causes of multiple neurodevelopmental disorders (*Stamberger et al., 2016*), and yet it remains unclear how its dysfunction causes disease.

Stxbp1/Munc18-1 is involved in synaptic vesicle docking, priming, and fusion through multiple interactions with the neuronal soluble *N*-ethylmaleimide-sensitive factor-attachment protein receptors (SNAREs) (*Rizo and Xu, 2015*). Genetic deletion of Stxbp1 in worms, flies, mice, and fish abolishes neurotransmitter release and leads to lethality and cell-intrinsic degeneration of neurons (*Harrison et al., 1994*; *Verhage, 2000*; *Weimer et al., 2003*; *Heeroma et al., 2004*; *Grone et al., 2016*). In humans, *STXBP1* de novo heterozygous mutations cause several of the most severe forms of epileptic encephalopathies including Ohtahara syndrome (*Saitsu et al., 2008*; *Saitsu et al., 2010*), West syndrome (*Deprez et al., 2010*; *Otsuka et al., 2010*), Lennox-Gastaut syndrome (*Carvill et al., 2013*; *Allen et al., 2013*), Dravet syndrome (*Carvill et al., 2014*), and other types of early-onset epileptic encephalopathies (*Deprez et al., 2010*; *Mignot et al., 2011*; *Stamberger et al., 2016*). Furthermore, *STXBP1* is one of the most frequently mutated genes in sporadic intellectual disabilities and developmental disorders (*Hamdan et al., 2009*; *Hamdan et al., 2011*; *Rauch et al., 2012*; *Deciphering Developmental Disorders Study, 2015*; *Deciphering Developmental Disorders Study, 2017*; *Suri et al., 2017*). All *STXBP1* encephalopathy patients show intellectual disability, mostly severe to profound, and 95% of patients have epilepsy (*Stamberger et al., 2016*). More than 90% of patients have motor deficits, such as dystonia, spasticity, ataxia, hypotonia, and tremor. Other clinical features in subsets of patients include developmental delay, hyperactivity, anxiety, stereotypies, aggressive behaviors, and autistic features (*Hamdan et al., 2009*; *Deprez et al., 2010*; *Mignot et al., 2011*; *Milh et al., 2011*; *Campbell et al., 2012*; *Rauch et al., 2012*; *Weckhuysen et al., 2013*; *Boutry-Kryza et al., 2015*; *Stamberger et al., 2016*; *Suri et al., 2017*).

*STXBP1* encephalopathy is mostly caused by haploinsufficiency because more than 60% of the reported mutations are either deletions, nonsense, frameshift, or splice site variants (*Stamberger et al., 2016*). A subset of missense variants were shown to destabilize the protein (*Saitsu et al., 2008*; *Saitsu et al., 2010*; *Guiberson et al., 2018*; *Kovacevic et al., 2018*) and cause aggregation to further reduce the wild type (WT) protein levels (*Guiberson et al., 2018*). Thus, partial loss-of-function of *Stxbp1* in vivo would offer opportunities to model *STXBP1* encephalopathy and study its pathogenesis. Indeed, removing *stxbp1b*, one of the two *STXBP1* homologs in zebrafish, caused spontaneous electrographic seizures (*Grone et al., 2016*). Three different *Stxbp1* null alleles have been generated in mice (*Verhage, 2000*; *Miyamoto et al., 2017*; *Kovacevic et al., 2018*). However, previous characterization of the corresponding heterozygous knockout mice was limited in scope, used relatively small cohorts, and yielded inconsistent results. For example, the reported cognitive phenotypes in mutant mice are mild or inconsistent between studies (*Miyamoto et al., 2017*; *Kovacevic et al., 2018*; *Orock et al., 2018*). Motor dysfunctions and several psychiatric deficits were not reported in previous studies (*Hager et al., 2014*; *Miyamoto et al., 2017*; *Kovacevic et al., 2018*; *Orock et al., 2018*). Thus, a comprehensive neurological and behavioral study of *Stxbp1* haploinsufficiency models is still lacking. Interestingly, Stxbp1 protein levels were reduced by only 25% in the brain of one line of previous *Stxbp1* heterozygous knockout mice (*Orock et al., 2018*) and 25% in the cortex and 50% in the hippocampus of another line (*Miyamoto et al., 2017*). Although STXBP1 levels in human patients are unknown, mouse models with a stronger reduction in Stxbp1 levels are desirable to determine to what extent *Stxbp1* haploinsufficient mice can recapitulate the neurological phenotypes of *STXBP1* encephalopathy. Furthermore, it remains elusive how *STXBP1* haploinsufficiency in vivo leads to hyperexcitable neural circuits and neurological deficits.

To address these questions and enhance the robustness and reproducibility of preclinical models of *STXBP1* haploinsufficiency, we developed two new genetically distinct *Stxbp1* haploinsufficiency

mouse models and performed parallel studies on both of them. These mutant mice showed a 40–50% reduction of Stxbp1 protein levels in most brain regions and recapitulated all key phenotypes observed in the human condition including seizures and impairments in cognitive, psychiatric, and motor functions. Electrophysiological and optogenetic experiments revealed that *Stxbp1* haploinsufficiency reduced cortical inhibition through two distinct mechanisms from two main classes of GABAergic neurons: reducing the synaptic strength of parvalbumin-expressing (Pv) interneurons and decreasing the connectivity of somatostatin-expressing (Sst) interneurons. Thus, these results demonstrate a crucial role of *Stxbp1* in neurologic and psychiatric functions and indicate that *Stxbp1* haploinsufficient mice are construct- and face-valid models of *STXBP1* encephalopathy. Furthermore, the reduced inhibition is likely a major contributor to the cortical hyperexcitability and neurobehavioral phenotypes. The differential effects on Pv and Sst interneuron-mediated inhibition suggest synapse-specific functions of *Stxbp1* and also highlight the necessity of studying synaptic specificity and diversity in neural circuits of synaptopathies.

## Results

### Generation of two new genetically distinct *Stxbp1* haploinsufficiency mouse models

To model *STXBP1* haploinsufficiency in mice, we first generated a knockout-first (KO-first) allele (*tm1a*), in which the *Stxbp1* genomic locus was targeted with a multipurpose cassette (*Testa et al., 2004*; *Skarnes et al., 2011*). The targeted allele contains a splice acceptor site from *Engrailed 2* (*En2SA*), an encephalomyocarditis virus internal ribosomal entry site (*IRES*), *lacZ*, and SV40 polyadenylation element (*pA*) that trap the transcripts after exon 6, thereby truncating the *Stxbp1* mRNA. The trapping cassette (*En2SA-IRES-lacZ-pA*) and exon 7 are flanked by two *FRT* sites and two *loxP* sites, respectively (*Figure 1—figure supplement 1A*). By sequentially crossing with Flp and Cre germline deleter mice, we removed both the trapping cassette and exon 7 from the heterozygous KO-first mice, which leads to a premature stop codon in exon 8 and generates a conventional knockout (KO) allele (*tm1d*) (*Figure 1A*). Heterozygous KO (*Stxbp1$^{tm1d/+}$*) and KO-first (*Stxbp1$^{tm1a/+}$*) mice are maintained on the C57BL/6J isogenic background for all experiments.

Homozygous mutants (*Stxbp1$^{tm1d/tm1d}$* and *Stxbp1$^{tm1a/tm1a}$*) died immediately after birth because they were completely paralyzed and could not breathe, consistent with the previous *Stxbp1* null alleles (*Verhage, 2000*; *Miyamoto et al., 2017*). Western blots with antibodies recognizing either the N- or C-terminus of Stxbp1 showed that at embryonic day 17.5, Stxbp1 protein was absent in *Stxbp1$^{tm1d/tm1d}$* and *Stxbp1$^{tm1a/tm1a}$* mice, and reduced by 50% in *Stxbp1$^{tm1d/+}$* and *Stxbp1$^{tm1a/+}$* mice (*Figure 1—figure supplement 1B,C*), indicating that both *tm1d* and *tm1a* are null alleles. We surveyed the Stxbp1 protein levels in different brain regions of *Stxbp1$^{tm1d/+}$* and *Stxbp1$^{tm1a/+}$* mice at 3 months of age. Stxbp1 was reduced by 40–50% in most brain areas except the cerebellum and olfactory bulb where the reduction was 20–30% (*Figure 1B,C*). These results demonstrate that *Stxbp1$^{tm1d/+}$* and *Stxbp1$^{tm1a/+}$* are indeed *Stxbp1* haploinsufficient mice. In theory, the *tm1d* and *tm1a* alleles could produce a truncated Stxbp1 protein of 18 kD and 16 kD, respectively. However, no such truncated proteins were observed in either heterozygous or homozygous mutants (*Figure 1—figure supplement 1B*), most likely because the truncated *Stxbp1* transcripts were degraded due to nonsense-mediated mRNA decay (*Chang et al., 2007*).

### *Stxbp1* haploinsufficient mice show a reduction in survival and body weights, and developed hindlimb clasping

We bred *Stxbp1$^{tm1d/+}$* and *Stxbp1$^{tm1a/+}$* mice with WT mice and found that at the time of genotyping (i.e., around postnatal week 3) *Stxbp1$^{tm1d/+}$* and *Stxbp1$^{tm1a/+}$* mice are 40% and 43% of the total offspring, respectively (*Figure 1D* and *Figure 1—figure supplement 2A*), indicating a postnatal lethality phenotype. However, the lifespans of many mutant mice that survived through weaning were similar to those of WT littermates (*Figure 1—figure supplement 2B*). Thus, *Stxbp1* haploinsufficient mice show reduced survival, but this phenotype is not fully penetrant. *Stxbp1$^{tm1d/+}$* and *Stxbp1$^{tm1a/+}$* mice appeared smaller and their body weights were consistently about 20% less than their sex- and age-matched WT littermates (*Figure 1E,F*). At 4 weeks of age, *Stxbp1$^{tm1d/+}$* and *Stxbp1$^{tm1a/+}$* mice began to exhibit abnormal hindlimb clasping, indicative of dystonia or spasticity

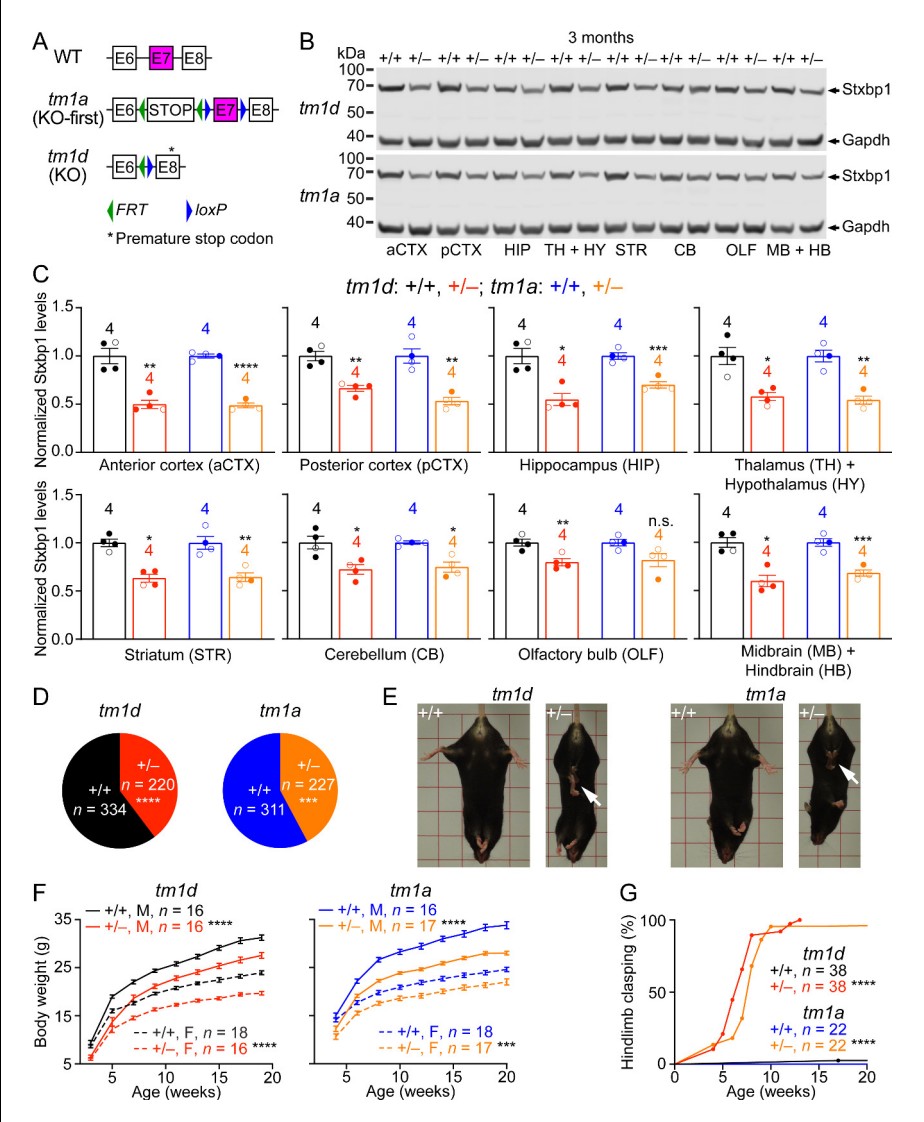

**Figure 1.** *Stxbp1* haploinsufficient mice exhibit reduced Stxbp1 levels, survival, and body weights and develop hindlimb clasping. (**A**) Genomic structures of *Stxbp1* WT, *tm1a* (KO-first), and *tm1d* (KO) alleles. In the *tm1a* allele, the STOP including the *En2SA-IRES-lacZ-pA* trapping cassette (see *Figure 1—figure supplement 1A*) truncates the *Stxbp1* mRNA after exon 6. In the *tm1d* allele, exon 7 is deleted, resulting in a premature stop codon in exon 8. E, exon; *FRT*, Flp recombination site; *loxP*, Cre recombination site. (**B**) Representative Western blots of proteins from different brain regions of 3-month-old WT, *Stxbp1^tm1d/+^*, and *Stxbp1^tm1a/+^* mice. Gapdh, a housekeeping protein as loading control. The brain regions are labeled by the same abbreviations as in (**C**). (**C**) Summary data of normalized Stxbp1 expression levels from different brain regions. Stxbp1 levels were first normalized by the Gapdh levels and then by the average Stxbp1 levels of all WT mice from the same blot. Each filled (male) or open (female) circle represents one mouse. (**D**) *Stxbp1^tm1d/+^* and *Stxbp1^tm1a/+^* male mice were crossed with WT female mice. Pie charts show the observed genotypes of the offspring at weaning (i.e., around the age of 3 weeks). *Stxbp1^tm1d/+^* and *Stxbp1^tm1a/+^* mice were significantly less than Mendelian expectations. (**E**) *Stxbp1^tm1d/+^* and *Stxbp1^tm1a/+^* mice were smaller and showed hindlimb clasping (arrows). (**F**) Body weights as a function of age. M, male; F, female. (**G**) The fraction of mice with hindlimb clasping as a function of age. Bar graphs are mean ± s.e.m. **, p<0.01; ***, p<0.001; ****, p<0.0001.

The online version of this article includes the following figure supplement(s) for figure 1:

**Figure supplement 1.** Generation of two new *Stxbp1* null alleles.

**Figure supplement 2.** Reduced survival of *Stxbp1* haploinsufficient mice.

(*Figure 1E*). By the age of 3 months, almost all mutant mice developed hindlimb clasping (*Figure 1G*). Thus, these observations indicate neurological deficits in *Stxbp1* haploinsufficient mice.

Guided by the symptoms of *STXBP1* encephalopathy human patients, we sought to perform behavioral and physiological assays to further examine the neurologic and psychiatric functions in male and female *Stxbp1* haploinsufficient mice and their sex- and age-matched WT littermates.

## Impaired motor and normal sensory functions in *Stxbp1* haploinsufficient mice

Motor impairments including dystonia, spasticity, ataxia, hypotonia, and tremor are frequently observed in *STXBP1* encephalopathy patients. Thus, we first assessed general locomotion by the open-field test where a mouse is allowed to freely explore an arena (*Figure 2A*). The locomotion of *Stxbp1^tm1d/+* and *Stxbp1^tm1a/+* mice was largely normal, but they traveled longer distances and faster than WT mice, indicating that *Stxbp1* haploinsufficient mice are hyperactive (*Figure 2B,C*). Both *Stxbp1^tm1d/+* and *Stxbp1^tm1a/+* mice explored the center region of the arena less than WT mice (*Figure 2D*) and made less vertical movements (*Figure 2E*), indicating that the mutant mice are more anxious. This anxiety phenotype was later confirmed by two other assays that specifically assess anxiety (see below). We used a variety of assays to further evaluate motor functions. *Stxbp1* haploinsufficient mice performed similarly to WT mice in the rotarod test, dowel test, inverted screen test, and wire hang test (*Figure 2—figure supplement 1*). However, the forelimb grip strength of

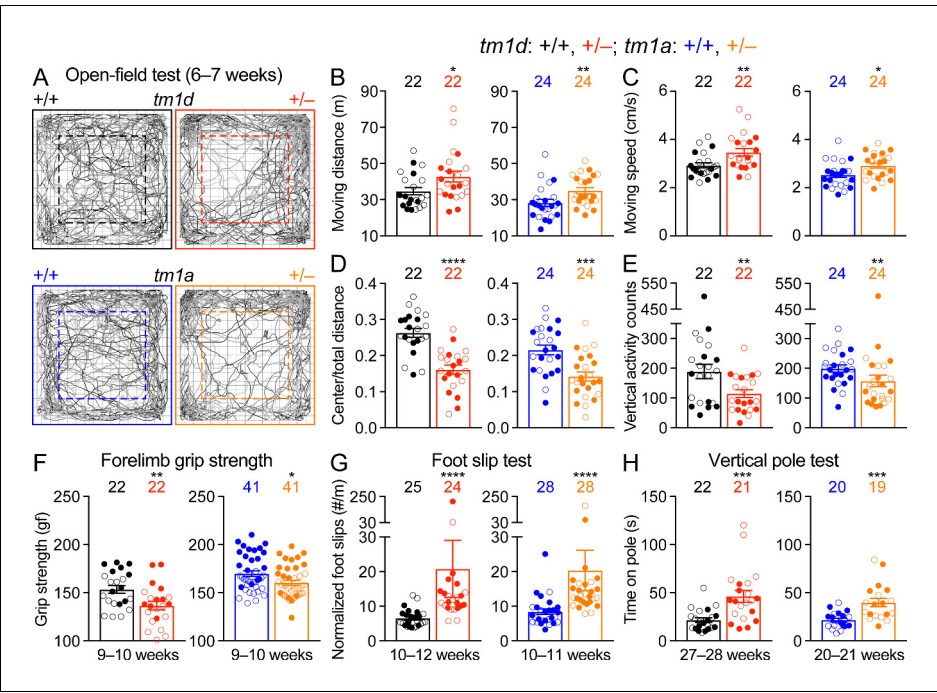

**Figure 2.** Motor dysfunctions of *Stxbp1* haploinsufficient mice. (**A**) Representative tracking plots of the mouse positions in the open-field test. Note that *Stxbp1^tm1d/+* and *Stxbp1^tm1a/+* mice traveled less in the center (dashed box) than WT mice. (**B–E**) Summary data showing hyperactivity and anxiety-like behaviors of *Stxbp1^tm1d/+* and *Stxbp1^tm1a/+* mice in the open-field test. *Stxbp1^tm1d/+* and *Stxbp1^tm1a/+* mice showed an increase in the total moving distance (**B**) and speed (**C**), and a decrease in the ratio of center moving distances over total moving distance (**D**) and vertical activity (**E**). (**F–H**) *Stxbp1^tm1d/+* and *Stxbp1^tm1a/+* mice had weaker forelimb grip strength (**F**), made more foot slips per travel distance on a wire grid (**G**), and took more time to get down from a vertical pole (**H**). The numbers and ages of tested mice are indicated in the figures. Each filled (male) or open (female) circle represents one mouse. Bar graphs are mean ± s.e.m. *, p<0.05; **, p<0.01; ***, p<0.001; ****, p<0.0001. The online version of this article includes the following figure supplement(s) for figure 2:

**Figure supplement 1.** Normal performance of *Stxbp1^tm1d/+* mice in rotarod, dowel, inverted screen, and wire hang tests.

**Figure supplement 2.** *Stxbp1* haploinsufficient mice have normal sensory functions.

*Stxbp1* haploinsufficient mice was weaker (*Figure 2F*). Furthermore, in the foot slip test where a mouse is allowed to walk on a wire grid, both *Stxbp1tm1d/+* and *Stxbp1tm1a/+* mice were not able to place their paws precisely on the wire to hold themselves and made many more foot slips than WT mice (*Figure 2G*). To assess the agility of mice, we performed the vertical pole test, which is often used to measure the bradykinesia of parkinsonism. When mice were placed head-up on the top of a vertical pole, it took mutant mice longer to orient themselves downward and descend the pole than WT mice (*Figure 2H*). Together, these results indicate that *Stxbp1* haploinsufficient mice do not develop ataxia, but their fine motor coordination and muscle strength are reduced.

We next examined the acoustic sensory function and found that *Stxbp1tm1d/+* and *Stxbp1tm1a/+* mice showed normal startle responses to different levels of sound (*Figure 2—figure supplement 2A*). To test sensorimotor gating, we measured the pre-pulse inhibition where the startle response to a strong sound is reduced by a preceding weaker sound. *Stxbp1tm1d/+* and *Stxbp1tm1a/+* mice displayed similar pre-pulse inhibition as WT mice (*Figure 2—figure supplement 2B*). They also had normal nociception as measured by the hot plate test (*Figure 2—figure supplement 2C*). Thus, the sensory functions and sensorimotor gating of *Stxbp1* haploinsufficient mice are normal.

## Cognitive functions of *Stxbp1* haploinsufficient mice are severely impaired

Intellectual disability is a core feature of *STXBP1* encephalopathy, as the vast majority of patients have severe to profound intellectual disability (*Stamberger et al., 2016*). However, the learning and memory deficits described in the previous *Stxbp1* heterozygous knockout mice are mild and inconsistent (*Miyamoto et al., 2017*; *Kovacevic et al., 2018*; *Orock et al., 2018*). To assess cognitive functions, we tested *Stxbp1* haploinsufficient mice in three different paradigms, object recognition, associative learning and memory, and working memory. First, we performed the novel object recognition test that exploits the natural tendency of mice to explore novel objects to evaluate their memories. This task is thought to depend on the hippocampus and cortex (*Antunes and Biala, 2012*; *Cohen and Stackman, 2015*). When tested with an inter-trial interval of 24 hr, WT mice interacted more with the novel object than the familiar object, whereas *Stxbp1tm1d/+* and *Stxbp1tm1a/+* mice interacted equally between the familiar and novel objects (*Figure 3A*). We also evaluated *Stxbp1tm1d/+* mice with an inter-trial interval of 5 min and observed a similar deficit (*Figure 3—figure supplement 1A*). We noticed that mutant mice overall spent less time interacting with the objects than WT mice during the trials (*Figure 3—figure supplement 1B*), which might reduce their 'memory load' of the objects. We hence allowed *Stxbp1tm1d/+* mice to spend twice as much time as WT mice in each trial to increase their interaction time with the objects (*Figure 3—figure supplement 1C*), but they still showed a similar deficit in recognition memory (*Figure 3—figure supplement 1D*). Thus, both long-term and short-term recognition memories are impaired in *Stxbp1* haploinsufficient mice.

Second, we used the Pavlovian fear conditioning paradigm to evaluate associative learning and memory, in which a mouse learns to associate a specific environment (i.e., the context) and a sound (i.e., the cue) with electric foot shocks. The fear memory is manifested by the mouse freezing when it is subsequently exposed to this specific context or cue without electric shocks. At two tested ages, *Stxbp1tm1d/+* and *Stxbp1tm1a/+* mice displayed a profound reduction in both context- and cue-induced freezing behaviors when tested 24 hr after the conditioning paradigm (*Figure 3B–E*). We also tested *Stxbp1tm1d/+* mice 1 hr after the conditioning paradigm and observed similar deficits (*Figure 3—figure supplement 1E*). Since the acoustic startle response and nociception are intact in *Stxbp1* haploinsufficient mice (*Figure 2—figure supplement 2A,C*), these results indicate that *Stxbp1* haploinsufficiency impairs both hippocampus-dependent contextual and hippocampus-independent cued fear memories.

Finally, we used the Y maze spontaneous alternation test to examine working memory, but did not observe significant difference between *Stxbp1tm1d/+* and WT mice (*Figure 3—figure supplement 1F*). Taken together, our results indicate that both long-term and short-term forms of recognition and associative memories are severely impaired in *Stxbp1* haploinsufficiency mice, but their working memory is intact.

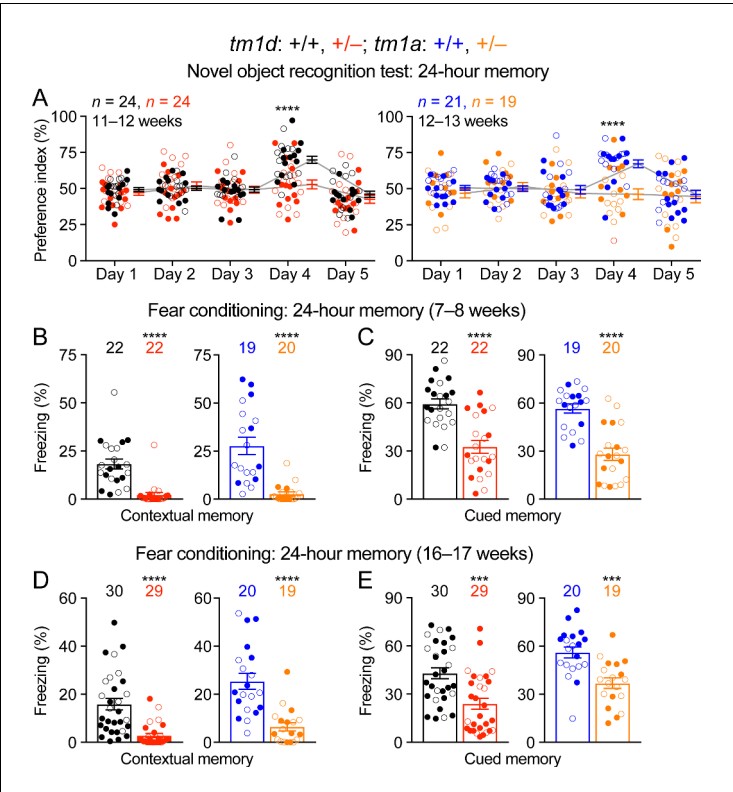

**Figure 3.** Impaired cognition of *Stxbp1* haploinsufficient mice. (**A**) In the novel object recognition test with 24 hr testing intervals, the ability of a mouse to recognize the novel object was assessed by the preference index (see Materials and methods). On days 1, 2, 3, and 5, mice were presented with the same two identical objects. In contrast to WT mice, *Stxbp1^{tm1d/+}* and *Stxbp1^{tm1a/+}* mice did not show a preference for the novel object on day 4 when they were presented with the familiar object and a novel object. (**B–E**) In the fear conditioning test, *Stxbp1^{tm1d/+}* and *Stxbp1^{tm1a/+}* mice at two different ages showed a reduction in both context-induced (**B,D**) and cue-induced (**C,E**) freezing behaviors 24 hr after training. The numbers and ages of tested mice are indicated in the figures. Each filled (male) or open (female) circle represents one mouse. Bar graphs are mean ± s.e.m. ***, p<0.001; ****, p<0.0001.

The online version of this article includes the following figure supplement(s) for figure 3:

**Figure supplement 1.** *Stxbp1* haploinsufficient mice show an impairment in object recognition and fear memory, but not working memory.

## *Stxbp1* haploinsufficient mice exhibit an increase in anxiety-like and repetitive behaviors

A number of psychiatric phenotypes including hyperactivity, anxiety, stereotypies, aggression, and autistic features were reported in subsets of *STXBP1* encephalopathy patients. We used a battery of behavioral assays to characterize each of these features in *Stxbp1* haploinsufficiency mice. The open-field test indicates that *Stxbp1* haploinsufficiency mice are hyperactive and more anxious than WT mice (*Figure 2A–E*). To specifically assess anxiety-like behaviors, we tested *Stxbp1^{tm1d/+}* and *Stxbp1^{tm1a/+}* mice in the elevated plus maze and light-dark chamber tests where a mouse is allowed to explore the open or closed arms of the maze and the clear or black chamber of the box, respectively. *Stxbp1^{tm1d/+}* and *Stxbp1^{tm1a/+}* mice entered the open arms and clear chamber less frequently and traveled shorter distance in the open arms and clear chamber than WT mice (*Figure 4A–D*; *Figure 4—figure supplement 1A–D*). Hence, these results confirm the heightened anxiety in *Stxbp1* haploinsufficient mice and are consistent with the previous studies (*Hager et al., 2014*; *Miyamoto et al., 2017*; *Kovacevic et al., 2018*).

To assess the stereotyped and repetitive behaviors, we used the hole-board test to measure the pattern of mouse exploratory nose poke (also called head dipping) behavior. As compared to WT

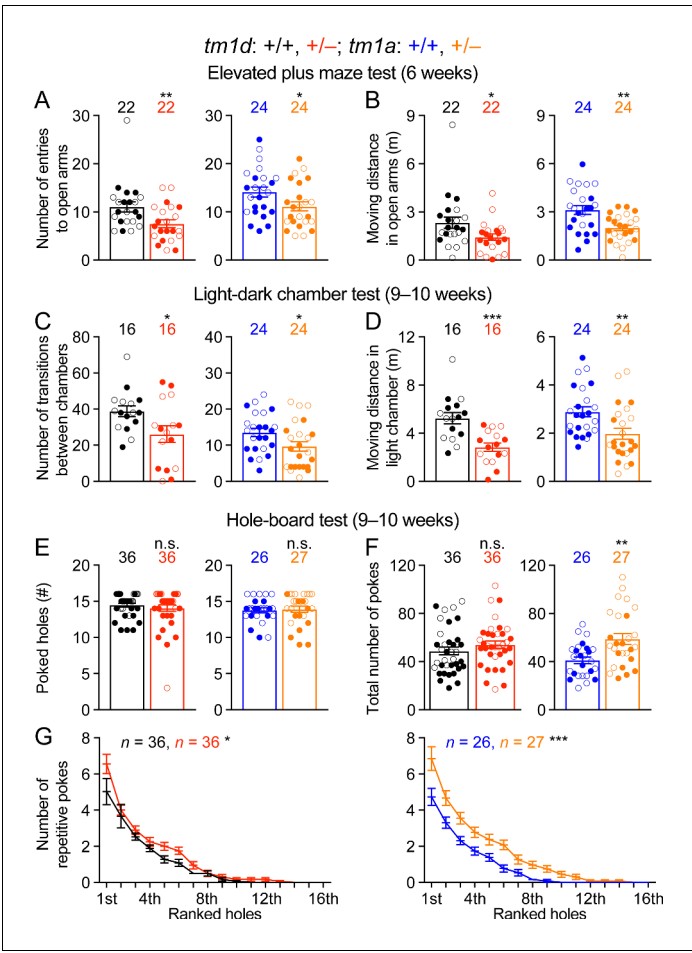

**Figure 4.** *Stxbp1* haploinsufficient mice show increased anxiety-like and repetitive behaviors. (**A,B**) In the elevated plus maze test, *Stxbp1*<sup>tm1d/+</sup> and *Stxbp1*<sup>tm1a/+</sup> mice entered the open arms less frequently (**A**) and traveled shorter distance in the open arms (**B**). (**C,D**) In the light-dark chamber test, *Stxbp1*<sup>tm1d/+</sup> and *Stxbp1*<sup>tm1a/+</sup> mice made less transitions between the light and dark chambers (**C**) and traveled shorter distance in the light chamber (**D**). (**E–G**) In the hole-board test, *Stxbp1*<sup>tm1d/+</sup> and *Stxbp1*<sup>tm1a/+</sup> mice poked similar numbers of holes as WT mice (**E**) and made similar or more total nose pokes (**F**). They made more repetitive nose pokes (i.e.,≥2 consecutive pokes) than WT mice across different holes (**G**). The numbers and ages of tested mice are indicated in the figures. Each filled (male) or open (female) circle represents one mouse. Bar graphs are mean ± s.e.m. n.s., p>0.05; *, p<0.05; **, p<0.01; ***, p<0.001.

The online version of this article includes the following figure supplement(s) for figure 4:

**Figure supplement 1.** The movements of *Stxbp1* haploinsufficient mice in elevated plus maze and light-dark chamber tests.

mice, *Stxbp1* haploinsufficient mice explored similar numbers of holes (*Figure 4E*) and made similar or larger numbers of nose pokes (*Figure 4F*). We analyzed the repetitive nose pokes (i.e.,≥2 consecutive pokes) into the same hole as a measure of repetitive behaviors. The mutant mice made more repetitive nose pokes than WT mice across many holes (*Figure 4G*), indicating that *Stxbp1* haploinsufficiency in mice causes abnormal stereotypy and repetitive behaviors, a psychiatric feature observed in about 20% of the *STXBP1* encephalopathy patients (*Stamberger et al., 2016*).

## Social aggression of *Stxbp1* haploinsufficient mice are elevated

During daily mouse husbandry, we noticed incidences of fighting and injuries of WT and *Stxbp1* haploinsufficient mice in their home cages, but no injuries were observed when *Stxbp1* haploinsufficient mice were singly housed, suggesting that the injuries likely resulted from fighting instead of self-injury. To formally examine aggressive behaviors, we first performed the resident-intruder test, in

which a male intruder mouse is introduced into the home cage of a male resident mouse, and the aggressive behaviors of the resident towards the intruder were scored. As compared to WT mice, male resident *Stxbp1^{tm1d/+}* and *Stxbp1^{tm1a/+}* mice were more likely to attack and spent more time attacking the intruders (*Figure 5A–C*). Another paradigm to assess aggression and social dominance is the tube test, in which two mice are released into the opposite ends of a tube, and the more dominant and aggressive mouse will win the competition by pushing its opponent out of the tube. When *Stxbp1^{tm1d/+}* and *Stxbp1^{tm1a/+}* mice were placed against their sex- and age-matched WT littermates, *Stxbp1* haploinsufficient mice won more competitions despite their smaller body sizes (*Figure 5D*). Thus, *Stxbp1* haploinsufficiency elevates innate aggression in mice.

To further evaluate social interaction, we performed the three-chamber test where a mouse is allowed to interact with an object or a sex- and age-matched partner mouse. Like WT mice, *Stxbp1^{tm1d/+}* and *Stxbp1^{tm1a/+}* mice preferred to interact with the partner mice rather than the objects (*Figure 5E*), indicating that *Stxbp1* haploinsufficiency does not compromise general sociability. Interestingly, the mutant mice in fact spent significantly more time than WT mice interacting with the partner mice ($p<0.0001$ for *Stxbp1^{tm1d/+}* vs. WT and $p=0.0015$ for *Stxbp1^{tm1a/+}* vs. WT), which might be due to the increased aggression of the mutant mice. Furthermore, we used the partition

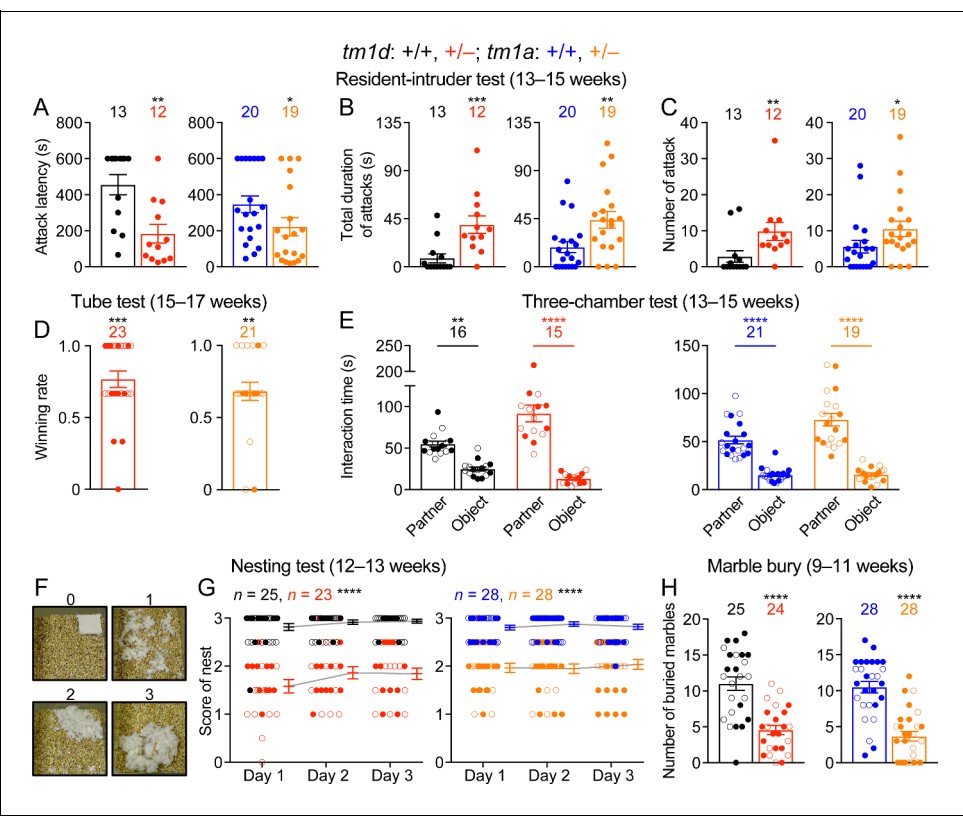

**Figure 5.** *Stxbp1* haploinsufficient mice show increased aggressive behaviors and reduced nest building and digging behaviors. (**A–C**) In the resident-intruder test, male *Stxbp1^{tm1d/+}* and *Stxbp1^{tm1a/+}* mice showed a reduction in the latency to attack the male intruder mice (**A**). The total duration (**B**) and number (**C**) of their attacks were increased as compared to WT mice. (**D**) In the tube test, *Stxbp1^{tm1d/+}* and *Stxbp1^{tm1a/+}* mice won more competitions against their WT littermates. (**E**) In the three-chamber test, *Stxbp1^{tm1d/+}* and *Stxbp1^{tm1a/+}* mice showed a preference in interacting with the partner mouse over the object. (**F,G**) *Stxbp1^{tm1d/+}* and *Stxbp1^{tm1a/+}* mice built poor quality nests. The quality of the nests was scored according to the criteria in (**F**) for three consecutive days (**G**). (**H**) *Stxbp1^{tm1d/+}* and *Stxbp1^{tm1a/+}* mice buried fewer marbles than WT mice. The numbers and ages of tested mice are indicated in the figures. Each filled (male) or open (female) circle represents one mouse. Bar graphs are mean ± s.e.m. *, $p<0.05$; **, $p<0.01$; ***, $p<0.001$; ****, $p<0.0001$.

The online version of this article includes the following figure supplement(s) for figure 5:

**Figure supplement 1.** *Stxbp1* haploinsufficient mice show normal social interactions.

test to examine the preference for social novelty, in which a mouse is allowed to interact with a familiar or novel partner mouse. Both WT and $Stxbp1^{tm1d/+}$ mice preferentially interacted more with the novel partner mice (*Figure 5—figure supplement 1A*). These results indicate that the general sociability and interest in social novelty are normal in *Stxbp1* haploinsufficient mice.

## Reduced nest building and digging behaviors in *Stxbp1* haploinsufficient mice

To further assess the well-being and psychiatric phenotypes of *Stxbp1* haploinsufficient mice, we performed the Nestlet shredding test and marble burying test to examine two innate behaviors, nest building and digging, respectively. We provided a Nestlet (pressed cotton square) to each mouse in the home cage and scored the degree of shredding and nest quality after 24, 48, and 72 hr (*Figure 5F*). $Stxbp1^{tm1d/+}$ and $Stxbp1^{tm1a/+}$ mice consistently scored lower than WT mice at all time points (*Figure 5G*). In the marble burying test, the $Stxbp1^{tm1d/+}$ and $Stxbp1^{tm1a/+}$ mice buried fewer marbles than WT mice (*Figure 5H*). The interpretation of marble burying remains controversial, as it may measure anxiety, compulsive-like behavior, or simply digging behavior (*Deacon, 2006*; *Thomas et al., 2009*; *Wolmarans et al., 2016*). Since *Stxbp1* haploinsufficient mice show elevated anxiety and repetitive behaviors, the reduced marble burying likely reflects an impairment of digging behavior, possibly due to the motor deficits. Likewise, the motor deficits may also contribute to the reduced nest building behavior.

## Cortical hyperexcitability and epileptic seizures in *Stxbp1* haploinsufficient mice

Another core feature of *STXBP1* encephalopathy is epilepsy with a broad spectrum of seizure types, such as epileptic spasm, focal, tonic, clonic, myoclonic, and absence seizures (*Stamberger et al., 2016*; *Suri et al., 2017*). To investigate if *Stxbp1* haploinsufficient mice have abnormal cortical activity and epileptic seizures, we performed chronic video-electroencephalography (EEG) and electromyography (EMG) recordings in freely moving $Stxbp1^{tm1d/+}$ mice and their sex- and age-matched WT littermates. We implanted three EEG electrodes in the frontal and somatosensory cortices and an EMG electrode in the neck muscles to record intracranial EEG and EMG, respectively, for at least 72 hr (*Figure 6A*). The phenotypes of each mouse are summarized in *Supplementary file 1*. $Stxbp1^{tm1d/+}$ mice exhibited cortical hyperexcitability and several epileptiform activities. First, they had numerous spike-wave discharges (SWDs) that typically were 3–6 Hz and lasted 1–2 s (*Figure 6C, E,F*). These oscillations showed similar characteristics to those generalized spike-wave discharges observed in animal models of absence seizures (*Maheshwari and Noebels, 2014*; *Depaulis and Charpier, 2018*). A much smaller number of SWDs with similar characteristics were also observed in WT mice (*Figure 6B,E*), consistent with previous studies (*Arain et al., 2012*; *Letts et al., 2014*). On average, the frequency of SWD episodes in $Stxbp1^{tm1d/+}$ mice was more than 40-fold higher than that in WT mice (*Figure 6E,F*). Importantly, SWDs frequently occurred in a cluster manner (i.e.,$\geq$5 episodes with an inter-episode-interval of $\leq$60 s) in $Stxbp1^{tm1d/+}$ mice, which never occurred in WT mice (*Figure 6—figure supplement 1*; *Video 1*). Furthermore, 56 episodes of SWDs from 10 out of 13 $Stxbp1^{tm1d/+}$ mice lasted more than 4 s, among which 54 episodes occurred during rapid eye movement (REM) sleep (*Figure 6D*; *Video 2*) and the other two episodes occurred when mice were awake. In contrast, only 1 out of 11 WT mice had 3 episodes of such long SWDs, all of which occurred when mice were awake (*Supplementary file 1*). In $Stxbp1^{tm1d/+}$ mice, SWDs were most frequent during the night, but occurred throughout the day and night (*Figure 6F*), indicating a general cortical hyperexcitability and abnormal synchrony in *Stxbp1* haploinsufficient mice.

Second, $Stxbp1^{tm1d/+}$ mice experienced frequent myoclonic seizures that manifested as sudden jumps or more subtle, involuntary muscle jerks associated with EEG discharges (*Figure 6G,H*). The large movement artifacts associated with the myoclonic jumps precluded proper interpretation of EEG signals, but this type of myoclonic seizures was observed in all 13 recorded $Stxbp1^{tm1d/+}$ mice and the majority of episodes occurred during REM or non-rapid eye movement (NREM) sleep (*Figure 6I*; *Video 3*). There were three similar jumps in 2 out of 11 WT mice that were indistinguishable from those in $Stxbp1^{tm1d/+}$ mice, but all of them occurred when mice were awake (*Figure 6I*). Moreover, the more subtle myoclonic jerks occurred frequently and often in clusters in $Stxbp1^{tm1d/+}$ mice, whereas only isolated events were observed in WT mice at a much lower frequency

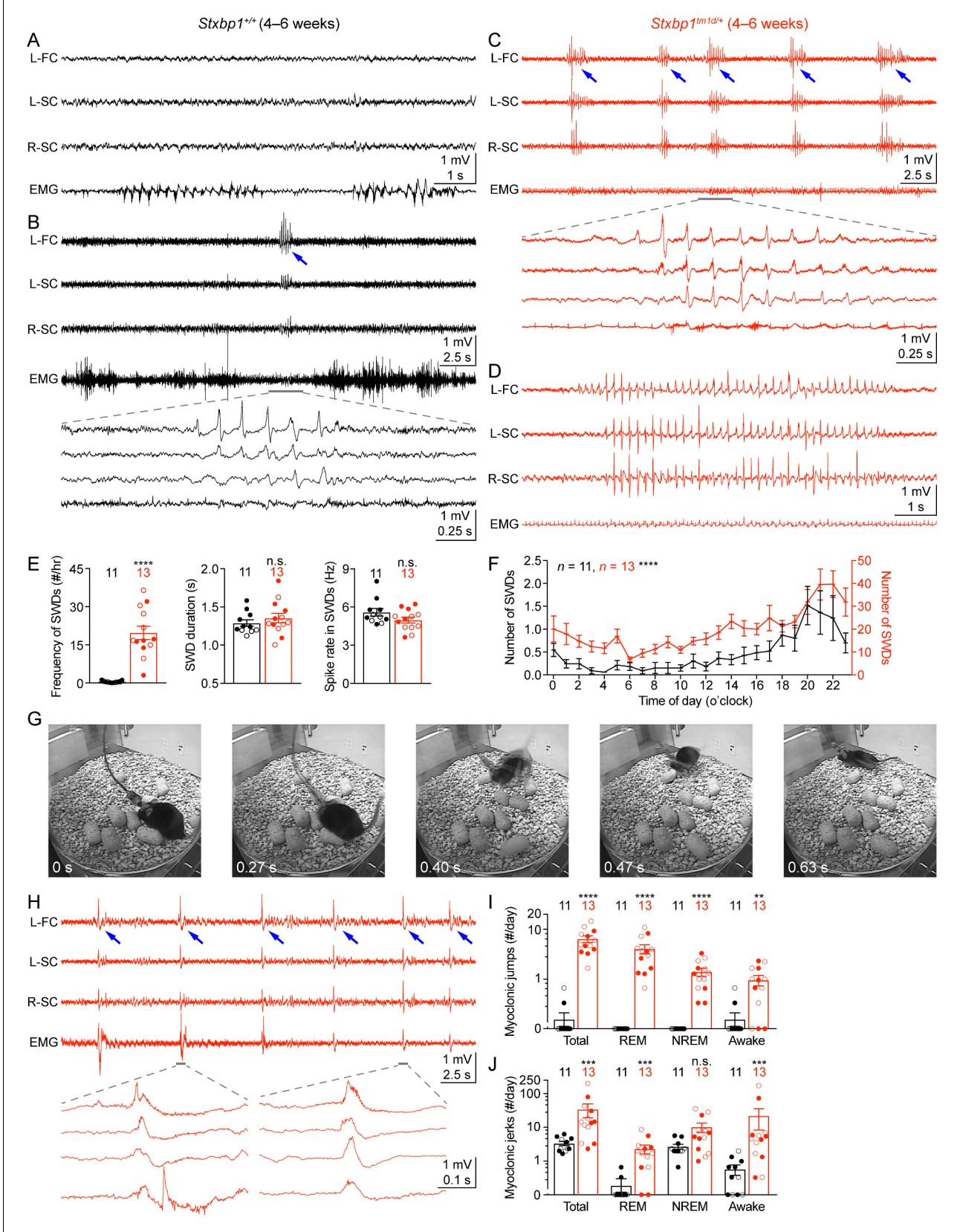

**Figure 6.** *Stxbp1tm1d/+* mice exhibit cortical hyperexcitability and epileptic seizures. (**A–D**) Representative EEG traces of the left frontal cortex (L-FC), left somatosensory cortex (L-SC), and right somatosensory cortex (R-SC), and EMG traces of the neck muscle from WT (**A,B**) and *Stxbp1tm1d/+* mice (**C,D**).
*Figure 6 continued on next page*

*Figure 6 continued*

Spike-wave discharges (SWDs, indicated by the blue arrows) occurred frequently and often in a cluster manner in *Stxbp1^{tm1d/+}* mice (see **Video 1**). The gray line-highlighted SWDs from WT and *Stxbp1^{tm1d/+}* mice were expanded to show the details of the oscillations (B,C). A long SWD (i.e.,>4 s) during REM sleep from a *Stxbp1^{tm1d/+}* mouse is shown in (D) (see **Video 2**). (E) Summary data showing the overall SWD frequency (left panel), duration (middle panel), and average spike rate (right panel). (F) The numbers of SWDs per hour in WT (left Y axis) and *Stxbp1^{tm1d/+}* (right Y axis) mice are plotted as a function of time of day and averaged over 3 days. (G) Video frames showing a myoclonic jump from a *Stxbp1^{tm1d/+}* mouse (see **Video 3**). The mouse was in REM sleep before the jump. (H) Representative EEG and EMG traces showing myoclonic jerks (indicated by the blue arrows) from a *Stxbp1^{tm1d/+}* mouse (see **Video 4**). Two episodes of myoclonic jerks highlighted by the gray lines were expanded to show that the EEG discharges occurred prior to (the first episode) or simultaneously with (the second episode) the EMG discharges. (I,J) Summary data showing the frequencies of two types of myoclonic seizures in different behavioral states. The numbers and ages of recorded mice are indicated in the figures. Each filled (male) or open (female) circle represents one mouse. Bar graphs are mean ± s.e.m. n.s., p>0.05; **, p<0.01; ***, p<0.001; ****, p<0.0001.

The online version of this article includes the following figure supplement(s) for figure 6:

**Figure supplement 1.** The clustering of SWDs in *Stxbp1^{tm1d/+}* mice does not result from a random distribution of frequent SWD episodes.

---

(*Figure 6H,J*; *Video 4*). EEG and EMG recordings showed that the cortical EEG spikes associated with the myoclonic jerks occurred before or simultaneously with the neck muscle EMG discharges (*Figure 6H*), consistent with the cortical or subcortical origins of myoclonuses, respectively (*Avanzini et al., 2016*).

## Normal cortical neuron densities in *Stxbp1* haploinsufficient mice

To identify cellular mechanisms that may underlie the cortical hyperexcitability and neurological deficits in *Stxbp1* haploinsufficient mice, we first examined the general cytoarchitecture and neuronal densities in the somatosensory cortex, as Stxbp1 affects neuronal survival and migration (*Verhage, 2000*; *Hamada et al., 2017*). Immunostaining of a pan-neuronal marker NeuN revealed a grossly normal cytoarchitecture and cortical lamination in adult *Stxbp1^{tm1d/+}* mice (*Figure 7A,B*). The densities of cortical neurons and two major classes of inhibitory neurons, Pv and Sst interneurons, were similar between *Stxbp1^{tm1d/+}* and WT mice (*Figure 7B–D*). Thus, *Stxbp1* haploinsufficiency does not appear to affect cortical neuron survival and migration.

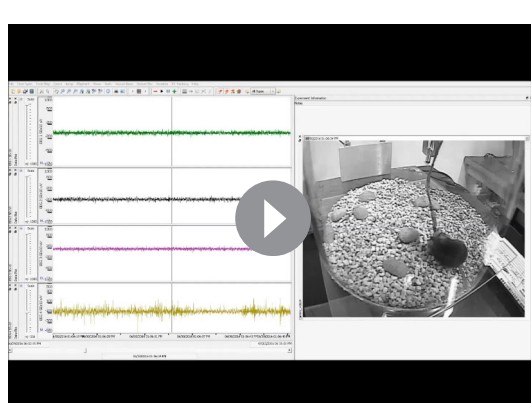

**Video 1.** *Stxbp1^{tm1d/+}* mice show clusters of SWDs. A representative video showing a SWD cluster in a *Stxbp1^{tm1d/+}* mouse. The top three traces are EEG signals from the left frontal cortex, right somatosensory cortex, and left somatosensory cortex. The bottom trace is the EMG signal from the neck muscle. The vertical line indicates the time of the current video frame. Note that the EEG signal from the left somatosensory cortex (the third channel) is inverted.

https://elifesciences.org/articles/48705#video1

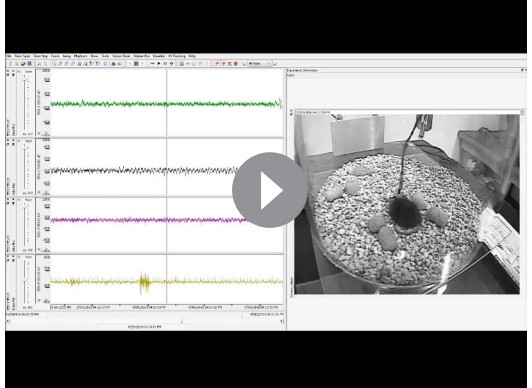

**Video 2.** *Stxbp1^{tm1d/+}* mice show long SWDs. A representative video showing a long SWD during REM sleep in a *Stxbp1^{tm1d/+}* mouse. The top three traces are EEG signals from the left frontal cortex, right somatosensory cortex, and left somatosensory cortex. The bottom trace is the EMG signal from the neck muscle. The vertical line indicates the time of the current video frame. Note that the EEG signal from the left somatosensory cortex (the third channel) is inverted.

https://elifesciences.org/articles/48705#video2

## *Stxbp1* haploinsufficiency reduces cortical inhibition in a synapse-specific manner

We next examined neuronal excitability and synaptic transmission in the somatosensory cortex. Whole-cell current clamp recordings of layer 2/3 pyramidal neurons in acute brain slices revealed only a small increase in the input resistances of $Stxbp1^{tm1d/+}$ neurons as compared to WT neurons (*Figure 8—figure supplement 1*). Previous studies showed that synaptic transmission was reduced in the cultured hippocampal neurons from heterozygous *Stxbp1* knockout mice and human neurons derived from heterozygous *STXBP1* knockout embryonic stem cells (*Toonen et al., 2006*; *Patzke et al., 2015*; *Orock et al., 2018*). However, such a decrease in excitatory transmission onto excitatory neurons is probably inadequate to explain how *Stxbp1* haploinsufficiency in vivo leads to cortical hyperexcitability. Genetic deletion of one copy of *Stxbp1* from GABAergic neurons led to early lethality in a subset of mice, suggesting a crucial role of Stxbp1 in GABAergic neurons *Kovacevic et al. (2018)*, but see *Miyamoto et al. (2017)* and *Miyamoto et al. (2019)*. Thus, we focused on the inhibitory synaptic transmission originating from Pv and Sst interneurons. A Cre-dependent tdTomato reporter line, *Rosa26-CAG-LSL-tdTomato* (*Madisen et al., 2010*), and *Pv-ires-Cre* (*Hippenmeyer et al., 2005*) or *Sst-ires-Cre* (*Taniguchi et al., 2011*) were used to identify Pv or Sst interneurons, respectively. We used whole-cell current clamp to stimulate a single Pv or Sst interneuron in layer 2/3 with a brief train of action potentials and whole-cell voltage clamp to record the resulting unitary inhibitory postsynaptic currents (uIPSCs) in a nearby pyramidal neuron (*Figure 8A, E*). The connectivity rate of Pv interneurons to pyramidal neurons was unaltered in $Stxbp1^{tm1d/+}$;$Rosa26^{tdTomato/+}$;$Pv^{Cre/+}$ mice (*Figure 8B*), but the unitary connection strength was reduced by 45% as compared to $Stxbp1^{+/+}$;$Rosa26^{tdTomato/+}$;$Pv^{Cre/+}$ mice (*Figure 8C*). In contrast, $Stxbp1^{tm1d/+}$;$Rosa26^{tdTomato/+}$;$Sst^{Cre/+}$ mice showed a 26% reduction in the connectivity rate of Sst interneurons to pyramidal neurons (*Figure 8F*), but the unitary connection strength was normal (*Figure 8G*). The short-term synaptic depression of both inhibitory connections during the train of stimulations was normal (*Figure 8D,H*). The inter-soma distances of interneurons and pyramidal neurons were similar between WT and mutant mice (Pv: WT 35.2 ± 2.4 μm, $n$ = 33, mutant 33.2 ± 2.5 μm, $n$ = 31, p=0.69; Sst: WT 31.4 ± 2.4 μm, $n$ = 36, mutant 32.0 ± 2.1 μm, $n$ = 36, p=0.65). Furthermore, we recorded the spontaneous excitatory postsynaptic currents (sEPSCs) in Pv and Sst interneurons and did not observe any significant changes of either amplitude or frequency in the mutant mice (*Figure 8—*

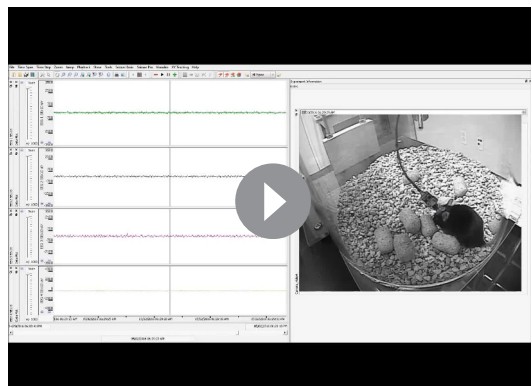

**Video 3.** $Stxbp1^{tm1d/+}$ mice show myoclonic jumps. A representative video showing a myoclonic jump of a $Stxbp1^{tm1d/+}$ mouse. The top three traces are EEG signals from the left frontal cortex, right somatosensory cortex, and left somatosensory cortex. The bottom trace is the EMG signal from the neck muscle. The vertical line indicates the time of the current video frame. Note that the EEG signal from the left somatosensory cortex (the third channel) is inverted.
https://elifesciences.org/articles/48705#video3

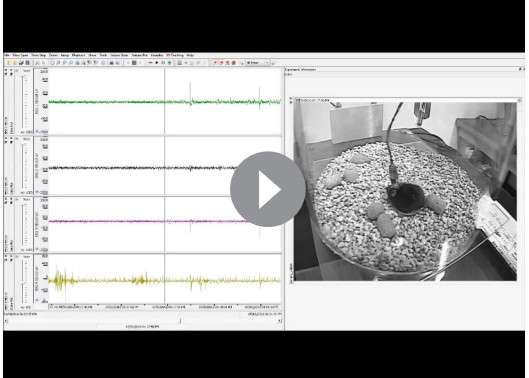

**Video 4.** $Stxbp1^{tm1d/+}$ mice show myoclonic jerks. A representative video showing a myoclonic jerk of a $Stxbp1^{tm1d/+}$ mouse. The top three traces are EEG signals from the left frontal cortex, right somatosensory cortex, and left somatosensory cortex. The bottom trace is the EMG signal from the neck muscle. The vertical line indicates the time of the current video frame. Note that the EEG signal from the left somatosensory cortex (the third channel) is inverted.
https://elifesciences.org/articles/48705#video4

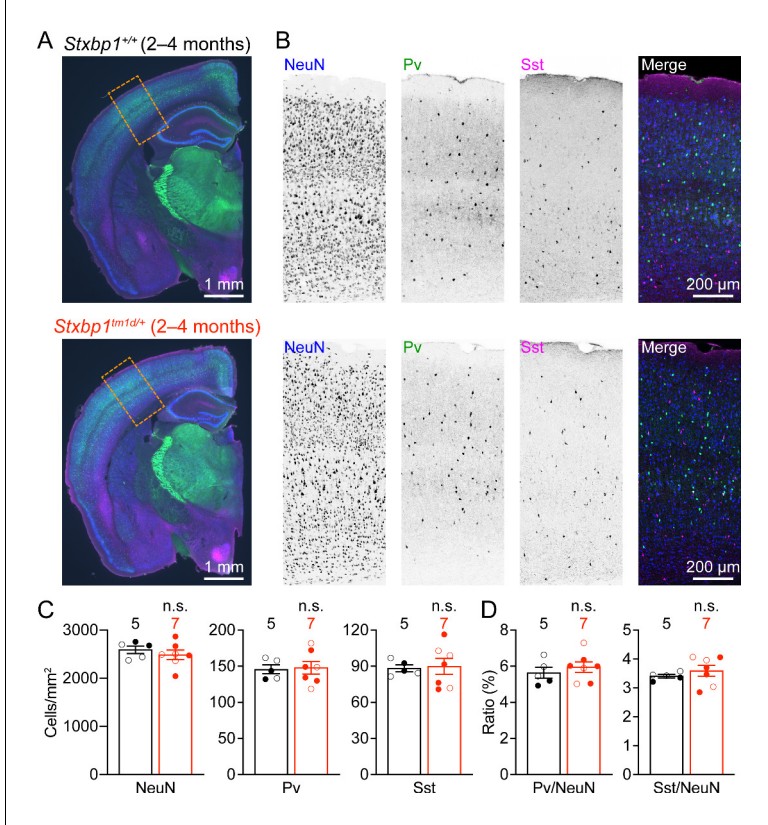

**Figure 7.** Cortical neuron densities are unaltered in *Stxbp1*^{tm1d/+}^ mice. (**A**) Representative fluorescent images of coronal sections stained by antibodies against NeuN (blue), Pv (green), and Sst (magenta). Note the similar cytoarchitecture between WT (upper panel) and *Stxbp1*^{tm1d/+}^ (lower panel) mice. (**B**) Representative fluorescent images of the somatosensory cortices within the boxed regions in (**A**) for WT (upper panels) and *Stxbp1*^{tm1d/+}^ (lower panels) mice. (**C**) Summary data showing similar densities of neurons (i.e., NeuN positive cells), Pv, and Sst interneurons in the somatosensory cortices of WT and *Stxbp1*^{tm1d/+}^ mice. (**D**) Summary data showing that the ratios of Pv and Sst interneurons to all somatosensory cortical neurons are similar between WT and *Stxbp1*^{tm1d/+}^ mice. The numbers and ages of mice are indicated in the figures. Each filled (male) or open (female) circle represents one mouse. Bar graphs are mean ± s.e.m. n.s., p>0.05.

*figure supplement 2*), suggesting that the excitatory drive onto interneurons is normal in *Stxbp1* haploinsufficient mice.

To determine the properties of quantal inhibitory transmission, we developed a new optogenetic method to isolate quantal IPSCs mediated by the GABA release specifically from Pv or Sst interneurons. We expressed a blue light-gated cation channel, channelrhodopsin-2 (ChR2) (*Nagel et al., 2003*; *Boyden et al., 2005*; *Li et al., 2005*), in Pv interneurons by injecting a Cre recombinase-dependent adeno-associated virus (AAV) into the somatosensory cortices of *Stxbp1*^{tm1d/+}^;*Pv*^{Cre/+}^ and *Stxbp1*^{+/+}^;*Pv*^{Cre/+}^ mice (*Figure 9A*). We recorded miniature IPSCs (mIPSCs) from layer 2/3 pyramidal neurons in the presence of voltage-gated sodium channel blocker, tetrodotoxin (TTX), but without any voltage-gated potassium channel blockers. Under such conditions light activation of ChR2 did not evoke synchronous neurotransmitter release, but could enhance asynchronous exocytosis of synaptic vesicles from Pv interneurons, resulting in an increase in the frequency of mIPSCs (*Figure 9B*). We also replaced the extracellular Ca^{2+}^ with Sr^{2+}^ to further reduce the likelihood of synchronous release. Instead of using a constant light intensity, we gradually decreased the photostimulation strength to minimize the tonic currents (*Figure 9—figure supplement 1*). We mathematically subtracted the mIPSCs recorded during the baseline period (i.e., before blue light stimulation) from those recorded during blue light stimulation to obtain the average amplitude, charge, and decay time constant of Pv interneurons-mediated quantal IPSCs (*Figure 9B*), which were all similar between *Stxbp1*^{tm1d/+}^;*Pv*^{Cre/+}^ and *Stxbp1*^{+/+}^;*Pv*^{Cre/+}^ mice (*Figure 9E*). Using this optogenetic method, we

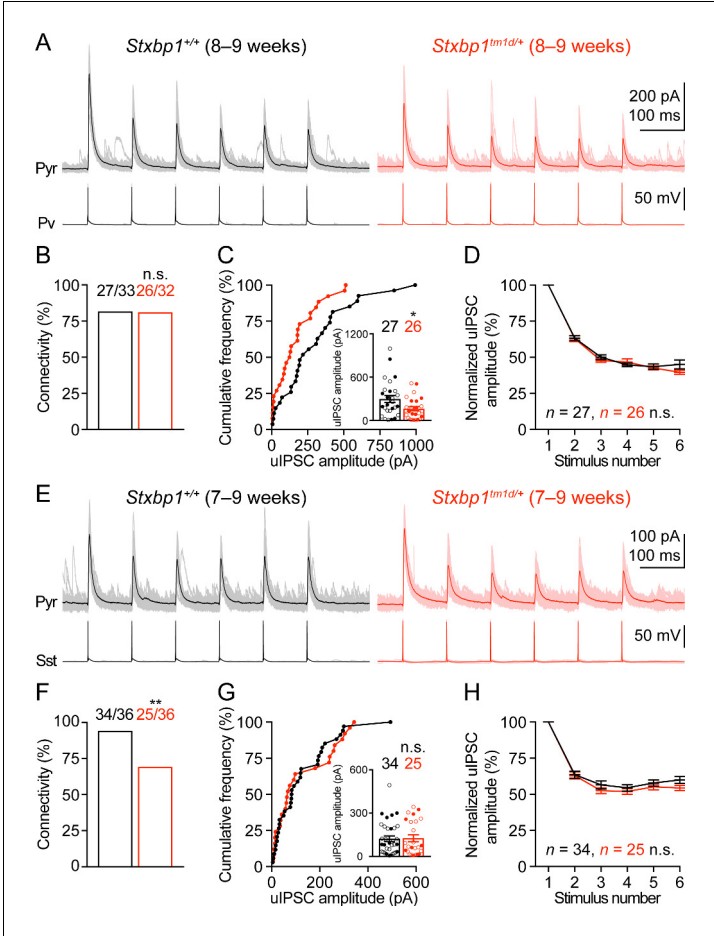

**Figure 8.** Inhibitory synapses from Pv and Sst interneurons are differentially impaired in *Stxbp1*$^{tm1d/+}$ mice. (**A**) uIPSCs of a layer 2/3 pyramidal neuron ($V_m$ = + 10 mV) in the somatosensory cortex (upper panels) evoked by a train of 10 Hz action potentials in a nearby Pv interneuron (lower panels) from WT and *Stxbp1*$^{tm1d/+}$ mice. 50 individual traces (lighter color) and the average trace (darker color) are superimposed. Note smaller uIPSCs in the *Stxbp1*$^{tm1d/+}$ neuron. (**B**) Unitary connectivity rates from Pv interneurons to pyramidal neurons were similar between WT (27 connections out of 33 pairs) and *Stxbp1*$^{tm1d/+}$ (26 connections out of 32 pairs) mice. (**C**) Cumulative frequencies of uIPSC amplitudes evoked by the first action potentials in the trains (median: WT, 217.3 pA; *Stxbp1*$^{tm1d/+}$, 127.1 pA). Inset, each filled (male) or open (female) circle represents the uIPSC amplitude of one synaptic connection. (**D**) uIPSC amplitudes during the trains of action potentials were normalized by the amplitudes of the first uIPSCs. Note the similar synaptic depression between WT and *Stxbp1*$^{tm1d/+}$ neurons. (**E–H**) Similar to (A–D), but for Sst interneurons. Unitary connectivity rates from Sst interneurons to pyramidal neurons (**F**) in *Stxbp1*$^{tm1d/+}$ mice (25 connections out of 36 pairs) were less than WT mice (34 connections out of 36 pairs). The uIPSC amplitudes evoked by the first action potentials in the trains (**G**, median: 83.5 pA and 68.0 pA, respectively) and synaptic depression (**H**) were similar between WT and *Stxbp1*$^{tm1d/+}$ mice. The ages of mice are indicated in the figures. Bar graphs are mean ± s.e.m. n.s., p>0.05; \*, p<0.05; \*\*, p<0.01.

The online version of this article includes the following figure supplement(s) for figure 8:

**Figure supplement 1.** Intrinsic neuronal excitability of *Stxbp1*$^{tm1d/+}$ mice is slightly increased.

**Figure supplement 2.** Spontaneous excitatory inputs onto Pv and Sst interneurons are unaltered in *Stxbp1*$^{tm1d/+}$ mice.

---

also found that the average amplitude, charge, and decay time constant of quantal IPSCs mediated by Sst interneurons were normal in *Stxbp1*$^{tm1d/+}$;*Sst*$^{Cre/+}$ mice (*Figure 9C,D,F*). Thus, *Stxbp1* haploinsufficiency does not affect the postsynaptic properties of inhibitory transmission.

Altogether, our results indicate that the reduction in the strength of Pv interneuron synapses is most likely due to a decrease in the number of readily releasable vesicles or release probability because the quantal amplitude and connectivity are unaltered in *Stxbp1*$^{tm1d/+}$ mice. Since Sst

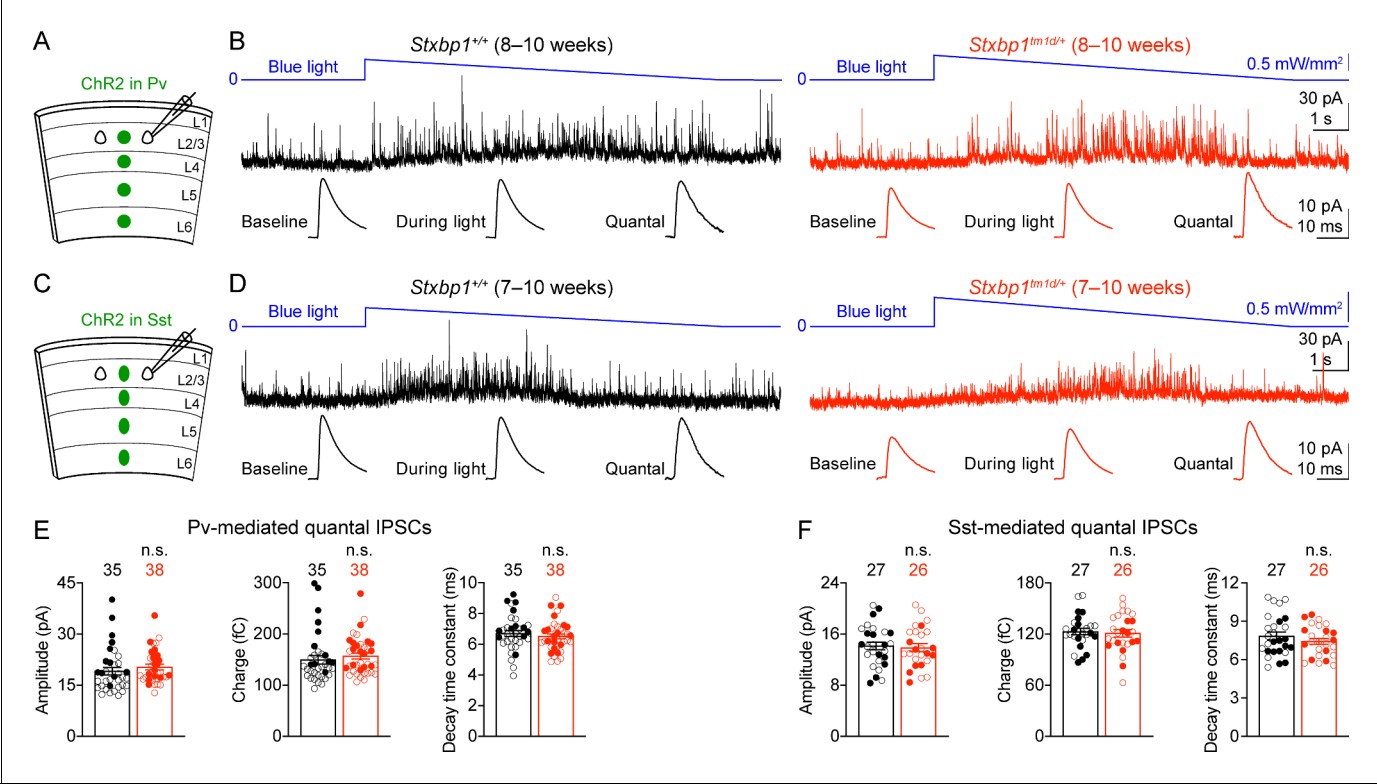

**Figure 9.** Pv and Sst interneurons-mediated quantal IPSCs are isolated by a novel optogenetic method and are unaltered in *Stxbp1*[tm1d/+] mice. (**A**) Schematic of slice experiments in (**B**). ChR2 in Pv interneurons. (**B**) mIPSCs in a layer 2/3 pyramidal neuron ($V_m$ = + 10 mV) from the somatosensory cortex of WT or *Stxbp1*[tm1d/+] mice. The intensity of blue light is indicated above the mIPSC traces. Note the increase of mIPSC frequency during blue light stimulation. The quantal IPSC trace was computed by subtracting the average mIPSC trace of the baseline period from that of the light stimulation period (bottom row). (**C,D**) As in (**A,B**), but for ChR2 in Sst interneurons. (**E,F**) Summary data showing that the average amplitude, charge, and decay time constant of Pv (**E**) or Sst (**F**) interneuron-mediated quantal IPSCs are similar between WT and *Stxbp1*[tm1d/+] mice. The numbers and ages of recorded neurons are indicated in the figures. Each filled (male) or open (female) circle represents one neuron. Bar graphs are mean ± s.e.m. n.s., p>0.05.

The online version of this article includes the following figure supplement(s) for figure 9:

**Figure supplement 1.** Ramping down blue light intensity minimizes the tonic currents during optogenetic activation of interneurons.

interneuron density and overall neuron density are normal in *Stxbp1*[tm1d/+] mice (**Figure 7**), a reduction in the connectivity rate of Sst interneurons to pyramidal neurons suggests a decrease in the number of inhibitory inputs onto pyramidal neurons. Thus, cortical inhibition mediated by both Pv and Sst interneurons is impaired in *Stxbp1* haploinsufficient mice, representing a likely cellular mechanism for the cortical hyperexcitability, seizures, and neurobehavioral deficits.

## Discussion

Extensive biochemical and structural studies of Stxbp1/Munc18-1 have elucidated its crucial role in synaptic vesicle exocytosis (**Rizo and Xu, 2015**), but provided little insight into its functional role at the organism level. Hence, apart from being an essential gene, the significance of *STXBP1* dysfunction in vivo was not appreciated until its de novo heterozygous mutations were discovered first in epileptic encephalopathies (**Saitsu et al., 2008**) and later in other neurodevelopmental disorders (**Hamdan et al., 2009**; **Hamdan et al., 2011**; **Rauch et al., 2012**; **Deciphering Developmental Disorders Study, 2015**). In this study, we generated two new lines of *Stxbp1* haploinsufficient mice (*Stxbp1*[tm1d/+] and *Stxbp1*[tm1a/+]) and systematically characterized them in all of the neurologic and psychiatric domains affected by *STXBP1* encephalopathy. These mice exhibit reduced survival, hindlimb clasping, impaired motor coordination, learning and memory deficits, hyperactivity, increased anxiety-like and repetitive behaviors, aggression, and epileptic seizures. Sensory abnormality has not

been documented in *STXBP1* encephalopathy patients (*Stamberger et al., 2016*) and we also did not observe any sensory dysfunctions in *Stxbp1* haploinsufficient mice. Thus, despite the large phenotypic spectrum of *STXBP1* encephalopathy in humans, our *Stxbp1* haploinsufficient mice recapitulate all key features of this neurodevelopmental disorder and are construct and face valid models of *STXBP1* encephalopathy. Importantly, the identical phenotypes of $Stxbp1^{tm1d/+}$ and $Stxbp1^{tm1a/+}$ mice demonstrate the robustness and reproducibility of these preclinical models, providing a foundation to further study the disease pathogenesis and explore therapeutic strategies. About 17% of the *STXBP1* encephalopathy patients showed autistic traits (*Stamberger et al., 2016*), but we and others (*Miyamoto et al., 2017*; *Kovacevic et al., 2018*) did not observe an impairment of social interaction in mutant mice using the three-chamber and partition tests. Perhaps the elevated aggression in *Stxbp1* haploinsufficient mice confounds these tests, or new mouse models that more precisely mimic the genetic alterations in the subset of *STXBP1* encephalopathy patients with autistic features are required to recapitulate this social behavioral phenotype.

Prior studies using the other three lines of *Stxbp1* heterozygous knockout mouse models reported only a subset of the neurologic and psychiatric deficits that we observed here (*Supplementary file 2*). For example, the reduced survival, hindlimb clasping, motor dysfunction, and increased repetitive behavior were not documented in the previous models. The previously reported cognitive phenotypes were much milder than what we observed. Both $Stxbp1^{tm1d/+}$ and $Stxbp1^{tm1a/+}$ mice showed severe impairments in the novel objection recognition and fear conditioning tests. In contrast, another line of *Stxbp1* heterozygous knockout mice showed normal spatial learning in the Morris water maze and Barnes maze (a dry version of the spatial maze) in one study (*Kovacevic et al., 2018*), but reduced spatial learning and memory in the radial arm water maze in another study (*Orock et al., 2018*). Different behavioral tests could have contributed to such differences among studies. However, a subtle but perhaps key difference is the Stxbp1 protein levels in different lines of heterozygous mutant mice. Stxbp1 is reduced by 40–50% in most brain regions of our $Stxbp1^{tm1d/+}$ and $Stxbp1^{tm1a/+}$ mice, but only by 25–50% in previous heterozygous knockout mice (*Miyamoto et al., 2017*; *Orock et al., 2018*), which may lead to fewer or less severe phenotypes. Furthermore, our study utilized much larger cohorts of mice for phenotypic characterization than previous studies, which allowed us to more comprehensively detect neurologic and psychiatric deficits in $Stxbp1^{tm1d/+}$ and $Stxbp1^{tm1a/+}$ mice.

Dysfunction of cortical GABAergic inhibition has been widely considered as a primary defect in animal models of autism spectrum disorder, schizophrenia, Down syndrome, and epilepsy among other neurological disorders (*Ramamoorthi and Lin, 2011*; *Marín, 2012*; *Nelson and Valakh, 2015*; *Paz and Huguenard, 2015*; *Contestabile et al., 2017*; *Lee et al., 2017*). In many cases, the origins of GABAergic dysfunction were either unidentified or attributed to Pv interneurons. Sst interneurons have only been directly implicated in a few disease models (*Ito-Ishida et al., 2015*; *Rubinstein et al., 2015*) despite their important physiological functions. Here we identified distinct deficits at Pv and Sst interneuron synapses in *Stxbp1* haploinsufficient mice, suggesting that Stxbp1 may have diverse functions at distinct synapses. The reduction in the strength of Pv interneuron synapses is consistent with the previous results that basal synaptic transmission is reduced at the neuromuscular junctions of *Stxbp1* heterozygous null flies and mice (*Wu et al., 1998*; *Toonen et al., 2006*) and the glutamatergic synapses of human *STXBP1* heterozygous knockout neurons (*Patzke et al., 2015*). The reduced synaptic strength is likely due to a decrease in the number of readily releasable vesicles or release probability given the crucial role of Stxbp1 in synaptic vesicle priming and fusion (*Rizo and Xu, 2015*) and the fact that the quantal amplitude and connectivity are normal in $Stxbp1^{tm1d/+}$ mice. Although the short-term synaptic depression is unaltered in $Stxbp1^{tm1d/+}$ mice, a change in release probability is still possible because at the Pv interneuron synapses the short-term synaptic plasticity during a short train of action potentials is not sensitive to the release probability (*Kraushaar and Jonas, 2000*; *Luthi et al., 2001*). On the other hand, the reduction in the connectivity of Sst interneuron synapses is unexpected, as Stxbp1 has not yet been implicated in the formation or maintenance of synapses. Complete loss of Stxbp1 in mice does not appear to affect the initial formation of neural circuits, but causes cell-autonomous neurodegeneration and protein trafficking defects (*Verhage, 2000*; *Heeroma et al., 2004*; *Law et al., 2016*). Since Munc13-1/2 double knockout mice also lack synaptic exocytosis, but do not show neurodegeneration (*Varoqueaux et al., 2002*), the degeneration phenotype in Stxbp1 null mice is unlikely the result of total arrest of synaptic exocytosis. Thus, Stxbp1 may regulate other intracellular processes in

addition to presynaptic transmitter release, and we speculate that it may be involved in a protein trafficking process important for the formation or maintenance of Sst interneuron synapses. Future morphological and structural analyses of Sst interneuron synapses will be necessary to further confirm the involvement of Stxbp1 in synapse formation or maintenance. Nevertheless, the impairment of Pv and Sst interneuron-mediated inhibition likely constitutes a key mechanism underlying the cortical hyperexcitability and neurobehavioral phenotypes of *Stxbp1* haploinsufficient mice. Future studies using cell-type specific *Stxbp1* haploinsufficient mouse models will help determine the role of specific GABAergic interneurons in the disease pathogenesis.

There are over one hundred developmental brain disorders that arise from mutations in postsynaptic proteins (*Bayés et al., 2011*; *Deciphering Developmental Disorders Study, 2017*), whereas few neurodevelopmental disorders have been diagnosed with mutations in presynaptic proteins until recently. In addition to *STXBP1*, pathogenic variants in genes encoding other key components of the presynaptic neurotransmitter release machinery have been increasingly discovered in neurodevelopmental disorders. These include $Ca^{2+}$-sensor synaptotagmin 1 (*SYT1*), vesicle priming factor unc-13 homolog A (*UNC13A*), and all three components of the neuronal SNAREs, syntaxin 1B (*STX1B*), synaptosome associated protein 25 (*SNAP25*), and vesicle associated membrane protein 2 (*VAMP2*) (*Rohena et al., 2013*; *Schubert et al., 2014*; *Shen et al., 2014*; *Baker et al., 2015*; *Engel et al., 2016*; *Hamdan et al., 2017*; *Lipstein et al., 2017*; *Baker et al., 2018*; *Fukuda et al., 2018*; *Salpietro et al., 2019*; *Wolking et al., 2019*). Haploinsufficiency of these synaptic proteins is likely the leading disease mechanism because the majority of the cases were caused by heterozygous loss-of-function mutations. The clinical features of these disorders are diverse, but significantly overlap with those of *STXBP1* encephalopathy. The most common phenotypes are intellectual disability and epilepsy (or cortical hyperexcitability), which can be considered as the core features of these genetic synaptopathies. Thus, *Stxbp1* haploinsufficient mice are a valuable model to understand the cellular and circuit origins of these complex disorders and provide mechanistic insights into the growing list of neurodevelopmental disorders caused by synaptic dysfunction.

## Materials and methods

### Key resources table

| Reagent type (species) or resource | Designation | Source or reference | Identifiers | Additional information |
|---|---|---|---|---|
| Cell line (*M. musculus*) | *Stxbp1*<sup>tm1a(EUCOMM)Hmgu</sup> embryonic stem cell clones (C57BL/6N strain) | European Conditional Mouse Mutagenesis Program (EUCOMM) | HEPD0510_5_A09, HEPD0510_5_B10 | |
| Genetic reagent (*M. musculus*) | *Stxbp1*<sup>tm1a</sup> (C57BL/6J strain) | This paper | | |
| Genetic reagent (*M. musculus*) | *Stxbp1*<sup>tm1d</sup> (C57BL/6J strain) | This paper | | |
| Genetic reagent (*M. musculus*) | B6(Cg)-*Tyr*<sup>c-2J</sup>/J | The Jackson Laboratory | RRID:IMSR_JAX:000058 | |
| Genetic reagent (*M. musculus*) | *Rosa26-Flpo* (C57BL/6J strain) | The Jackson Laboratory | RRID:IMSR_JAX:012930 | |
| Genetic reagent (*M. musculus*) | *Sox2-Cre* (C57BL/6J strain) | The Jackson Laboratory | RRID:IMSR_JAX:008454 | |
| Genetic reagent (*M. musculus*) | C57BL/6J | The Jackson Laboratory | RRID:IMSR_JAX:000664 | |
| Genetic reagent (*M. musculus*) | *Pv-ires-Cre* (C57BL/6J strain) | The Jackson Laboratory | RRID:IMSR_JAX:017320 | |
| Genetic reagent (*M. musculus*) | *Sst-ires-Cre* (C57BL/6;129S4 strain) | The Jackson Laboratory | RRID:IMSR_JAX:013044 | |
| Genetic reagent (*M. musculus*) | *Rosa26-CAG-LSL-tdTomato* (C57BL/6J strain) | The Jackson Laboratory | RRID:IMSR_JAX:007914 | |
| Antibody | Rabbit anti-Munc18-1, polyclonal | Abcam, catalog # ab3451 | RRID:AB_303813 | (1:2000 or 1:5,000) |

*Continued on next page*

*Continued*

| Reagent type (species) or resource | Designation | Source or reference | Identifiers | Additional information |
|---|---|---|---|---|
| Antibody | Rabbit anti-Munc18-1, polyclonal | Synaptic Systems, catalog # 116002 | RRID:AB_887736 | (1:2000 or 1:5,000) |
| Antibody | Rabbit anti-Gapdh, polyclonal | Santa Cruz Biotechnology, catalog #sc-25778 | RRID:AB_10167668 | (1:300 or 1:1,000) |
| Antibody | Goat anti-rabbit IgG conjugated with IRDye 680LT, polyclonal | LI-COR Biosciences, catalog # 925–68021 | RRID:AB_2713919 | (1:20,000) |
| Antibody | Rabbit anti-Somatostatin, polyclonal | Peninsula Laboratories International, catalog # T4103.0050 | RRID:AB_518614 | (1:3,000) |
| Antibody | Mouse anti-Parvalbumin, monoclonal | EMD Millipore, catalog # MAB1572 | RRID:AB_2174013 | (1:1,000) |
| Antibody | Guinea pig anti-NeuN, polyclonal | Sigma Millipore, catalog # ABN90 | RRID:AB_11205592 | (1:1,000) |
| Antibody | Goat anti-guinea pig IgG (H+L) conjugated with Alexa Flour 488, polyclonal | Invitrogen, catalog # A-11073 | RRID:AB_2534117 | (1:1,000) |
| Antibody | Goat anti-mouse IgG (H+L) conjugated with Alexa Flour 555, polyclonal | Invitrogen, catalog # A-21424 | RRID:AB_141780 | (1:1,000) |
| Antibody | Goat anti-rabbit IgG (H+L) conjugated with Alexa Flour 647, polyclonal | Invitrogen, catalog # A-21245 | RRID:AB_141775 | (1:1,000) |
| Recombinant DNA reagent | pAAV-EF1α-DIO-hChR2 (H134R)-P2A-EYFP | This paper | Addgene: 139283 | This plasmid was used to produce the AAV vector used in *Figure 9*. |
| Transfected construct | AAV9-EF1α-DIO-hChR2 (H134R)-P2A-EYFP | This paper, Baylor College of Medicine Gene Vector Core | Addgene: 139283 | This AAV vector was used in *Figure 9*. |
| Software, algorithm | Axograph X 1.5.4 | AxoGraph | RRID:SCR_014284 | https://axograph.com |
| Software, algorithm | pClamp 10.7 | Molecular Devices | RRID:SCR_011323 | https://www.molecular devices.com |
| Software, algorithm | Image Studio Lite 5.0 | LI-COR Biosciences | RRID:SCR_013715 | https://www.licor.com |
| Software, algorithm | MATLAB R2015 to R2017 | MathWorks | RRID:SCR_001622 | https://www.mathworks.com |
| Software, algorithm | Prism 6.0, 7.0, and 8.0 | GraphPad | RRID:SCR_002798 | https://www.graphpad.com |
| Software, algorithm | Spyder 3.3.6 with Anaconda | Spyder | RRID:SCR_017585 | https://www.spyder-ide.org |
| Software, algorithm | Sirenia 1.7.2 to 1.8.3 | Pinnacle Technology | RRID:SCR_016183 | https://www.pinnaclet.com |
| Software, algorithm | Imaris 9.2 | Oxford Instruments | RRID:SCR_007370 | https://imaris.oxinst.com |

## Mice

*Stxbp1*[tm1a(EUCOMM)Hmgu] embryonic stem (ES) cell clones (C57Bl/6N strain) were obtained from the European Conditional Mouse Mutagenesis Program (EUCOMM) and the targeting was confirmed by Southern blots. Two ES cell clones (HEPD0510_5_A09 and HEPD0510_5_B10) were injected into blastocysts to generate chimeric mice. Germline transmission of clone HEPD0510_5_A09 was obtained by crossing chimeric mice to B6(Cg)-*Tyr*[c-2J]/J mice (JAX #000058) to establish the KO-first (*tm1a*) line. Heterozygous KO-first mice were crossed to *Rosa26-Flpo* mice (*Raymond and Soriano, 2007*) to remove the trapping cassette in the germline. The resulting offspring were then crossed to

*Sox2-Cre* mice (*Hayashi et al., 2002*) to delete exon seven in the germline to generate the KO (*tm1d*) line. Both *Rosa26-Flpo* and *Sox2-Cre* mice were obtained from the Jackson Laboratory (#012930 and 008454, respectively). *Stxbp1* mice were genotyped by PCR using primer sets 5'-TTCCACAGCCCTTTACAGAAAGG-3' and 5'-ATGTGTATGCCTGGACTCACAGGG-3' for WT allele, 5'-TTCCACAGCCCTTTACAGAAAGG-3' and 5'-CAACGGGTTCTTCTGTTAGTCC-3' for KO-first allele, and 5'-TTCCACAGCCCTTTACAGAAAGG-3' and 5'-TGAACTGATGGCGAGCTCAGACC-3' for KO allele.

Heterozygous *Stxbp1* KO-first and KO mice were crossed to wild type (WT) C57BL/6J mice (JAX #000664) for maintaining both lines on the C57BL/6J background and for generating experimental cohorts. Male BALB/cAnNTac mice were obtained from Taconic (#BALB-M). *Pv-ires-Cre* (*Hippenmeyer et al., 2005*), *Sst-ires-Cre* (*Taniguchi et al., 2011*), and *Rosa26-CAG-LSL-tdTomato* (*Madisen et al., 2010*) mice were obtained from the Jackson Laboratory (#017320, 013044, and 007914, respectively). *Pv-ires-Cre* and *Rosa26-CAG-LSL-tdTomato* mice were maintained on the C57BL/6J background. *Sst-ires-Cre* mice were on a C57BL/6;129S4 background. Heterozygous KO mice were crossed to *Rosa26-CAG-LSL-tdTomato* mice to generate $Stxbp1^{tm1d/+};Rosa26^{tdTomato/tdTomato}$ mice. *Pv-ires-Cre* and *Sst-ires-Cre* mice were then crossed to $Stxbp1^{tm1d/+};Rosa26^{tdTomato/tdTomato}$ mice to generate $Stxbp1^{tm1d/+};Rosa26^{tdTomato/+};Pv^{Cre/+}$ or $Stxbp1^{+/+};Rosa26^{tdTomato/+};Pv^{Cre/+}$ and $Stxbp1^{tm1d/+};Rosa26^{tdTomato/+};Sst^{Cre/+}$ or $Stxbp1^{+/+};Rosa26^{tdTomato/+};Sst^{Cre/+}$ mice, respectively. *Pv-ires-Cre* and *Sst-ires-Cre* mice were also crossed to $Stxbp1^{tm1d/+}$ mice to generate $Stxbp1^{tm1d/+};Pv^{Cre/+}$ or $Stxbp1^{+/+};Pv^{Cre/+}$ and $Stxbp1^{tm1d/+};Sst^{Cre/+}$ or $Stxbp1^{+/+};Sst^{Cre/+}$ mice, respectively.

Mice were housed in an Association for Assessment and Accreditation of Laboratory Animal Care International-certified animal facility on a 14 hr/10 hr light/dark cycle. All procedures to maintain and use mice were approved by the Institutional Animal Care and Use Committee at Baylor College of Medicine.

## Southern and Western blots

Southern and Western blot analyses were performed according standard protocols. For Southern blots, genomic DNA was extracted from ES cells and digested with BspHI for the 5' probe or MfeI for the 3' probe (*Figure 1—figure supplement 1A*). $^{32}$P-labeled probes were used to detect DNA fragments. For Western blots, proteins were extracted from the brains at embryonic day 17.5 or 3 months of age using lysis buffer containing 50 mM Tris-HCl (pH 7.6), 150 mM NaCl, 1 mM EDTA, 1% Triton X-100, 0.5% Na-deoxycholate, 0.1% SDS, and 1 tablet of cOmplete, Mini, EDTA-free Protease Inhibitor Cocktail (Roche) in 10 ml buffer. Stxbp1 was detected by a rabbit antibody against the N terminal residues 58–70 (Abcam, catalog # ab3451, lot # GR79394-18, 1:2000 or 1:5000 dilution) or a rabbit antibody against the C terminal residues 580–594 (Synaptic Systems, catalog # 116002, lot # 116002/15, 1:2000 or 1:5000 dilution). Gapdh was detected by a rabbit antibody (Santa Cruz Bio-technology, catalog # sc-25778, lot # A0515, 1:300 or 1:1000 dilution). Primary antibodies were detected by a goat anti-rabbit antibody conjugated with IRDye 680LT (LI-COR Biosciences, catalog # 925–68021, lot # C40917-01, 1:20,000 dilution). Proteins were visualized and quantified using an Odyssey CLx Imager and Image Studio Lite 5.0 (LI-COR Biosciences).

## Immunohistochemistry and fluorescent microscopy

Mice were anesthetized by an intraperitoneal injection of a ketamine and xylazine mix (80 mg/kg and 16 mg/kg, respectively) and transcardially perfused with phosphate buffered saline (PBS, pH 7.4) followed by 4% paraformaldehyde in PBS (pH 7.4). Brains were then post-fixed for 2 hr in 4% paraformaldehyde at 4˚C, cryoprotected with 30% sucrose, and sectioned into 50 µm coronal slices using a HM 450 Sliding Microtome (Thermo Scientific). Brain sections were stored in an ethylene gly-col:glycerol:PBS solution (1:1:1.3) until use. Sections containing the somatosensory cortex were incubated in blocking solution (0.2% Triton X-100 in PBS with 10% normal goat serum) for 2 hr and then with primary antibodies for 48 hr at 4˚C. Primary antibodies were diluted in the blocking solution: rabbit anti-Somatostatin (Peninsula Laboratories International, catalog # T4103.0050, lot # A17908, 1:3000), mouse anti-Parvalbumin (EMD Millipore, catalog # MAB1572, lot # 2982272, 1:1000), and guinea pig anti-NeuN (Sigma Millipore, catalog # ABN90, lot # 3253333, 1:1000). Sections were washed in 0.2% Triton X-100 in PBS and then incubated with the following secondary antibodies

diluted 1:1000 in blocking solution for 24 hr at 4°C: goat anti-guinea pig IgG (H+L) conjugated with Alexa Flour 488 (Invitrogen, catalog # A-11073, lot # 1841755), goat anti-mouse IgG (H+L) conjugated with Alexa Flour 555 (Invitrogen, catalog # A-21424, lot # 1588453), goat anti-rabbit IgG (H+L) conjugated with Alexa Flour 647 (Invitrogen, catalog # A-21245, lot # 1623067). Sections were washed in 0.2% Triton X-100 in PBS and mounted in ProLong Diamond Antifade Mountant with DAPI (Invitrogen, catalog # P36962).

Low magnification images of brain sections were acquired on an Axio Zoom.V16 Fluorescence Stereo Zoom Microscope (Zeiss). High magnification, tile scanned z-stack images of the primary somatosensory cortex were acquired on an Sp8X Confocal Microscope (Leica) using a 20 × oil objective. Three brain sections were imaged and quantified per mouse. Approximately 50 images were acquired per tile scan with a 5% overlap between images for tiling. The z-stack was centered in the middle of the brain section and 10 optical sections were taken at 0.39 μm step. For analysis, the three optical sections in the middle of the z-stack were processed using the 'Sum Slices' function in ImageJ (National Institutes of Health) and then the images were cropped to a region of approximately 2 mm$^2$ spanning all cortical layers. Within this region, each Pv or Sst interneuron was confirmed to be co-labeled with DAPI and NeuN and counted manually. The numbers of NeuN positive cells were estimated using the Surfaces function in Imaris 9.2 (Oxford Instruments) with the following parameters: surface grain size = 0.568 μm, eliminating background of largest sphere = 9 μm diameter, threshold = 30, seed point diameter = 7 μm, seed point quality = 10, and number of voxels < 200. Accuracy of surface detection was verified by manually counting NeuN positive cells in images containing about 200 cells and the error rate was less than 10%.

## DNA construct, AAV production, and injection

Plasmid pAAV-EF1α-DIO-hChR2(H134R)-P2A-EYFP was generated by replacing the hChR2(C128A H134R) in pAAV-EF1α-DIO-hChR2(C128A H134R)-P2A-EYFP (*Prakash et al., 2012*) with the hChR2 (H134R) from pAAV-EF1α-DIO-hChR2(H134R)-EYFP (Addgene #20298) and was deposited at Addgene (#139283). The recombinant AAV vectors were produced by the Gene Vector Core at Baylor College of Medicine. To express ChR2 in Pv or Sst interneurons, 200 nl of AAV9-EF1α-DIO-hChR2 (H134R)-P2A-EYFP vectors (3 × 10$^{13}$ genome copies/ml) were injected into the somatosensory cortex of *Stxbp1$^{tm1d/+}$;Pv$^{Cre/+}$* and *Stxbp1$^{+/+}$;Pv$^{Cre/+}$* or *Stxbp1$^{tm1d/+}$;Sst$^{Cre/+}$* and *Stxbp1$^{+/+}$;Sst$^{Cre/+}$* mice, respectively, at postnatal day 1–5 as previously described (*Xue et al., 2014*; *Messier et al., 2018*) with an UltraMicroPump III and a Micro4 controller (World Precision Instruments).

## Behavioral tests

All behavioral experiments except the tube test were performed and analyzed blind to the genotypes. The numbers of mice needed were estimated based on previous studies using similar behavioral tests. Approximately equal numbers of *Stxbp1* mutant mice and their sex- and age-matched WT littermates of both sexes were tested in parallel in each experiment except the resident-intruder test where only male mice were used. In each cage, two mutant and two WT mice were housed together. Before all behavioral tests, mice were habituated in the behavioral test facility for at least 30 min. The sexes and ages of the tested mice were indicated in the figures.

### Open-field test

A mouse was placed in the center of a clear, open chamber (40 × 40 × 30 cm) and allowed to freely explore for 30 min in the presence of 700–750 lux illumination and 65 dB background white noise. In each chamber, two layers of light beams (16 for each layer) in the horizontal X and Y directions capture the locomotor activity of the mouse. The horizontal plane was evenly divided into 256 squares (16 × 16), and the center zone is defined as the central 100 squares (10 × 10). The horizontal travel and vertical activity were quantified by either an Open Field Locomotor system or a VersaMax system (OmniTech).

### Rotarod test

A mouse was placed on an accelerating rotarod apparatus (Ugo Basile). Each trial lasted for a maximum of 5 min, during which the rod accelerated linearly from 4 to 40 revolutions per minute (RPM) or 8 to 80 RPM. The time when the mouse walks on the rod and the latency for the mouse to fall

from the rod were recorded for each trial. Mice were tested in four trials per day for two consecutive days or in three trials per day for four consecutive days. There was a 30–60 min resting interval between trials.

### Dowel test
A mouse was placed in the center of a horizontal dowel (6.5 mm or 9.5 mm diameter) and the latency to fall was measured with a maximal cutoff time of 120 s.

### Inverted screen test
A mouse was placed onto a wire grid, and the grid was carefully picked up and shaken a couple of times to ensure that the mouse was holding on. The grid was then inverted such that the mouse was hanging upside down from the grid. The latency to fall was measured with a maximal cutoff time of 60 s.

### Wire hang test
A mouse was suspended by its forepaws on a 1.5 mm wire and the latency to fall was recorded with a maximal cutoff time of 60 s.

### Foot slip test
A mouse was placed onto an elevated 40 × 25 cm wire grid (1 × 1 cm spacing) and allowed to freely move for 5 min. The number of foot slips was manually counted, and the moving distance was measured through a video camera (ANY-maze, Stoelting). The number of foot slips were normalized by the moving distance for each mouse.

### Vertical pole test
A mouse was placed head-upward at the top of a vertical threaded metal pole (1.3 cm diameter, 55 cm length). The amount of time for the mouse to turn around and descend to the floor was measured with a maximal cutoff time of 120 s.

### Grip strength
Forelimb grip strength was measured using a Grip Strength Meter (Columbus Instruments). A mouse was held by the tail and allowed to grasp a trapeze bar with its forepaws. Once the mouse grasped the bar with both paws, the mouse was pulled away from the bar until the bar was released. The digital meter displayed the level of tension exerted on the bar in gram-force (gf).

### Acoustic startle response test
A mouse was placed in a well-ventilated, clear plastic cylinder and acclimated to the 70 dB background white noise for 5 min. The mouse was then tested with four blocks. Each block consisted of 13 trials, during which each of 13 different levels of sound (70, 74, 78, 82, 86, 90, 94, 98, 102, 106, 110, 114, or 118 dB, 40 ms, inter-trial interval of 15 s on average) was presented in a pseudorandom order. The startle response was recorded for 40 ms after the onset of the sound. The rapid force changes due to the startles were measured by an accelerometer (SR-LAB, San Diego Instruments).

### Pre-pulse inhibition test
A mouse was placed in a well-ventilated, clear plastic cylinder and acclimated to the 70 dB background noise for 5 min. The mouse was then tested with six blocks. Each block consisted of 8 trials in a pseudorandom order: a 'no stimulus' trial (40 ms, only 70 dB background noise present), a test pulse trial (40 ms, 120 dB), three different pre-pulse trials (20 ms, 74, 78, or 82 dB), and three different pre-pulse inhibition trials (a 20-ms, 74, 78, or 82-dB pre-pulse preceding a 40-ms, 120-dB test pulse by 100 milliseconds). The startle response was recorded for 40 ms after the onset of the 120 dB test pulse. The inter-trial interval is 15 s on average. The rapid force changes due to the startles were measured by an accelerometer (SR-LAB, San Diego Instruments). Pre-pulse inhibition of the startle responses was calculated as '1 – (pre-pulse inhibition trial/test pulse trial)'.

## Hot plate test

A mouse was placed on a hot plate (Columbus Instruments) with a constant temperature of 55℃. The latency for the mouse to first respond with either a hind paw lick, hind paw flick, or jump was measured. If the mouse did not respond within 45 s, then the test was terminated, and the latency was considered to be 45 s.

## Novel object recognition test

A mouse was first habituated in an empty arena (24 × 45 × 20 cm) for 5 min before every trial. The habituated mouse was then placed into the testing arena with two identical objects (i.e., familiar object 1 and familiar object 2) for the first three trials. In the fourth trial, familiar object 1 was replaced with a novel object.. In the fifth trial, the mouse was presented with the two original, identical objects again. Each trial lasted 5 min. The inter-trial interval was 24 hr or 5 min. In the modified version, $Stxbp1^{tm1d/+}$ and WT mice were exposed to the objects for 10 and 5 min during each trial, respectively. The movement of mice was recorded by a video camera placed above the test arena. The amount of time that the mouse interacted with the objects ($T$) was recorded using a wireless keyboard (ANY-maze, Stoelting). The preference index of interaction was calculated as $T_{familiar\ object\ 1}/(T_{familiar\ object\ 1} + T_{familiar\ object\ 2})$ for the first three trials and fifth trial and as $T_{novel\ object}/(T_{novel\ object} + T_{familiar\ object\ 2})$ for the fourth trial.

## Fear conditioning test

Pavlovian fear conditioning was conducted in a chamber (30 × 25 × 29 cm) that has a grid floor for delivering electrical shocks (Coulbourn Instruments). A camera above the chamber was used to monitor the mouse. During the 5 min training phase, a mouse was placed in the chamber for 2 min, and then a sound (85 dB, white noise) was turned on for 30 s immediately followed by a mild foot shock (2 s, 0.72 mA). The same sound and foot shock were repeated one more time 2 min after the first foot shock. After the second foot shock, the mouse stayed in the training chamber for 18 s before returning to its home cage. After 1 or 24 hr, the mouse was tested for the contextual and cued fear memories. In the contextual fear test, the mouse was placed in the same training chamber and its freezing behavior was monitored for 5 min without the sound stimulus. The mouse was then returned to its home cage. One to two hours later, the mouse was transferred to the chamber after it has been altered using plexiglass inserts and a different odor to create a new context for the cued fear test. After 3 min in the chamber, the same sound cue that was used in the training phase was turned on for 3 min without foot shocks while the freezing behavior was monitored. The freezing behavior was scored using an automated video-based system (FreezeFrame, Actimetrics). The freezing time (%) during the first 2 min of the training phase (i.e., before the first sound) was subtracted from the freezing time (%) during the contextual fear test. The freezing time (%) during the first 3 min of the cued fear test (i.e., without sound) was subtracted from the freezing time (%) during the last 3 min of the cued fear test (i.e., with sound).

## Y maze spontaneous alternation test

A mouse was placed in the center of a Y-shaped maze consisting of three walled arms (35 × 5 × 10 cm) and allowed to freely explore the different arms for 10 min. The sequence of the arms that the mouse entered was recorded using a video camera (ANY-maze, Stoelting). The correct choice refers to when the mouse entered an alternate arm after it came out of one arm.

## Elevated plus maze test

A mouse was placed in the center of an elevated maze consisting of two open arms (25 × 8 cm) and two closed arms with high walls (25 × 8 × 15 cm). The mouse was initially placed facing the open arms and then allowed to freely explore for 10 min in the presence of 700–750 lux illumination and 65 dB background white noise. The mouse activity was recorded using a video camera (ANY-maze, Stoelting).

## Light-dark chamber test

A mouse was placed in a rectangular light-dark chamber (44 × 21 × 21 cm) and allowed to freely explore for 10 min in the presence of 700–750 lux illumination and 65 dB background white noise.

One third of the chamber is made of black plexiglass (dark) and two thirds is made of clear plexiglass (light) with a small opening between the two areas. The movement of the mouse was tracked by the Open Field Locomotor system (OmniTech).

### Hole-board test

A mouse was placed at the center of a clear chamber (40 × 40 × 30 cm) that contains a black floor with 16 evenly spaced holes (5/8-inch diameter) arranged in a 4 × 4 array. The mouse was allowed to freely explore for 10 min. Its open-field activity above the floorboard and nose pokes into the holes were detected by infrared beams above and below the hole board, respectively, using the VersaMax system (OmniTech).

### Resident-intruder test

Male test mice (resident mice) were individually caged for 2 weeks before testing. Age-matched male white BALB/cAnNTac mice (Taconic) were group-housed to serve as the intruders. During the test, an intruder was placed into the home cage of a test mouse for 10 min and their behaviors were video recorded. Videos were scored for the number and duration of each attack by the resident mouse regardless the attack was initiated by either the resident or intruder.

### Tube test

A pair of a mutant mouse and an age- and sex-matched WT mouse that were housed in different home cages were placed into the opposite ends of a clear acrylic, cylindrical tube (3.5 cm diameter). The mouse that retreats backwards first was considered as the loser. The winner was scored as 1 and the loser as 0. Each mutant mouse was tested against three different WT mice and the scores were averaged.

### Three-chamber test

The apparatus (60.8 × 40.5 × 23 cm) consists of three chambers (left, center, and right) of equal size with 10 × 5 cm openings between the chambers. WT C57BL/6J mice were used as partner mice. A test mouse was placed in the apparatus with a mesh pencil cup in each of the left and right chambers and allowed to freely explore for 10 min. A novel object was then placed under one mesh pencil cup and an age- and sex-matched partner mouse under the other mesh pencil cup. The test mouse was allowed to freely explore for another 10 min. The position of the test mouse was tracked through a video camera (ANY-maze, Stoelting), and the approaches of the test mouse to the object or partner mouse were scored manually using a wireless keyboard. Partner mice were habituated to the mesh pencil cups in the apparatus for 1 hr per day for 2 days prior to testing. A partner mouse was used only in one test per day.

### Partition test

The partitioned cage is a standard mouse cage (28.5 × 17.5 × 12 cm) divided in half with a clear perforated partition (a hole of 0.6 cm diameter). WT C57BL/6J mice were used as partner mice. A test mouse was housed in one side of the partitioned cage for overnight. In the afternoon before testing, an age- and sex-matched partner mouse was placed in the opposite half of the partitioned cage. On the next day, the time and number of approaches of the test mouse to the partition were scored using a handheld Psion event recorder (Observer, Noldus) in three 5 min tests. The first test measured the approaches with the familiar overnight partner. The second measured the approaches with a novel partner mouse. The third test measured the approaches with the returned original partner mouse.

### Nestlet shredding test

A mouse was individually housed in its home cage and an autoclaved Nestlet was given to the mouse. The quality of the nest was assessed every 24 hr for three consecutive days.

## Marble burying test

A clean standard housing cage was filled with approximately 8 cm deep bedding material. 20 marbles were arranged on top of the bedding in a 4 × 5 array. A mouse was placed into this cage and remained undisturbed for 30 min before returning to its home cage. The number of buried marbles (i.e., at least 2/3 of the marble covered by the bedding) was recorded.

## Video-EEG/EMG

Mice at 3–4 weeks of age were anesthetized with 1.5–2.5% isoflurane in oxygen, and the body temperature was maintained by a feedback-based DC temperature control system at 37°C. The head was secured in a stereotaxic apparatus and an incision was made along the midline to expose the skull. Craniotomies (approximate diameter of 0.25 mm) were performed with a round bur (0.25 mm diameter) and a high-speed rotary micromotor at coordinates (see below) that were normalized by the distance between Bregma and Lambda (DBL). Perfluoroalkoxy polymer (PFA)-coated silver wire electrodes (127 µm bare diameter, 177.8 µm coated diameter, A-M Systems) were used for grounding, referencing, and recording. A grounding electrode was placed on the right frontal cortex. An EEG reference electrode was placed on the cerebellum. Three EEG electrodes were placed on the left frontal cortex (anterior posterior (AP): 0.42 of DBL, medial lateral (ML): 0.356 of DBL, dorsal ventral (DV): −1.5 mm), left, and right somatosensory cortices (AP: −0.34 of DBL, ML: ±0.653 of DBL, DV: −1.5 mm). An EMG recording and an EMG reference electrode were inserted into the neck muscles. All the electrodes were connected to an adapter that was secured on the skull by dental acrylic. The skin around the wound was sutured and mice were returned to the home cage to recover for at least one week. Before recording, mice were individually habituated in the recording chambers (10-inch diameter of plexiglass cylinder with bedding and access to food and water) for 24 hr. EEG/EMG signals were sampled at 5000 Hz with a 0.5 Hz high-pass filter and synchronous videos were recorded at 30 frames per second from freely moving mice for continuous 72 hr using a 4-channel EEG/EMG tethered system (Pinnacle Technology).

To detect spike-wave discharges (SWDs), EEG signals of each channel were divided into 10 min segments and each segment was filtered by a third order Butterworth bandpass filter with 0.5–400 Hz cutoffs. The filtered data was divided into 250-ms non-overlapping epochs. EEG signal changes that occurred in the time domain were captured by root mean square ($RMS = \sqrt{\sum_{i=1}^{i=n} s_i^2/n}$; $s$, EEG signal; $n$ = 1250) and spike density (number of spikes normalized to each epoch). EEG signal changes that occurred in the frequency domain were captured by frequency band ratio ($FBR = \sum_{n=f1}^{n=f2} ABS(FFT(n)) / \sum_{n=f3}^{n=f4} ABS(FFT(n))$; $f1$ = 100; $f2$ = 300; $f3$ = 0.5; $f4$ = 80) where the power of the upper band (100–300 Hz) was contrasted with that of the lower band (0.5–80 Hz). The above features were computed in MATLAB. An EEG segment that exceeded the thresholds for all of the above features was identified as a SWD candidate. The candidates were further classified by a convolutional neural network in Spyder (Spyder) that was trained with manually labeled EEG segments. The first layer of the network contained 32 filters that returned their matches with 10-ms (kernel size) non-overlapping (stride) candidate segments across the three EEG channels. Successive convolutional layers were stacked sequentially. For every two consecutive convolutional layers, there was a pooling layer that down-sampled the outputs by a factor of 5 to reduce computation. The overall network consisted of two layers of 32 filters, one layer of pooling, two layers of 64 filters, one layer of pooling, two layers of 128 filters, and one layer of pooling. The network was trained through an iterative approach. In each training iteration, the optimizer (Adadelta) updated the weights of the filters and the loss function (binary cross entropy) evaluated how well the network predicted SWDs. This iteration process continued until the loss function was minimized. Methods implemented to reduce overfitting included dropout (i.e., 50% of the neurons were randomly dropped out from calculation for each iteration) and early stopping (i.e., training process was stopped when the loss function on validation set did not decrease for three iterations). The trained neural network removed 99% of the false-positive candidates and the remaining candidates were further confirmed by visual inspection. For each SWD, the duration (the time difference between the first and last peaks) and spike rate were quantified. The SWD cluster was defined as a cluster of 5 or more SWD episodes that occurred with inter-episode-interval of maximal 60 s.

To identify myoclonic seizures, the EEG/EMG traces and videos were visually inspected to identify sudden jumps and myoclonic jerks. When the mouse suddenly and quickly move the body in less than one second, if one or more limbs leave the cage floor, then this is classified as a sudden jump. If all limbs stay on the cage floor, then this is classified as a myoclonic jerk. The state of the mouse right before the myoclonic seizure was classified as REM sleep, NREM sleep, or awake based on the EEG/EMG.

## Brain slice electrophysiology

All electrophysiological experiments were performed and analyzed blind to the genotypes. Mice were anesthetized by an intraperitoneal injection of a ketamine and xylazine mix (80 mg/kg and 16 mg/kg, respectively) and transcardially perfused with cold (0–4°C) slice cutting solution containing 80 mM NaCl, 2.5 mM KCl, 1.3 mM $NaH_2PO_4$, 26 mM $NaHCO_3$, 4 mM $MgCl_2$, 0.5 mM $CaCl_2$, 20 mM D-glucose, 75 mM sucrose and 0.5 mM sodium ascorbate (315 mosmol, pH 7.4, saturated with 95% $O_2$/5% $CO_2$). Brains were removed and sectioned in the cutting solution with a VT1200S vibratome (Leica) to obtain 300 μm coronal slices. Slices containing primary somatosensory cortex were collected and incubated in a custom-made interface holding chamber saturated with 95% $O_2$/5% $CO_2$ at 34°C for 30 min and then at room temperature for 20 min to 8 hr until they were transferred to the recording chamber.

Recordings were performed on submerged slices in artificial cerebrospinal fluid (ACSF) containing 119 mM NaCl, 2.5 mM KCl, 1.3 mM $NaH_2PO_4$, 26 mM $NaHCO_3$, 1.3 mM $MgCl_2$, 2.5 mM $CaCl_2$, 20 mM D-glucose and 0.5 mM sodium ascorbate (305 mosmol, pH 7.4, saturated with 95% $O_2$/5% $CO_2$, perfused at 3 ml/min) at 30–32°C. For whole-cell recordings, we used a $K^+$-based pipette solution containing 142 mM $K^+$-gluconate, 10 mM HEPES, 1 mM EGTA, 2.5 mM $MgCl_2$, 4 mM ATP-Mg, 0.3 mM GTP-Na, 10 mM $Na_2$-phosphocreatine (295 mosmol, pH 7.35) or a $Cs^+$-based pipette solution containing 121 mM $Cs^+$-methanesulfonate, 10 mM HEPES, 10 mM EGTA, 1.5 mM $MgCl_2$, 4 mM ATP-Mg, 0.3 mM GTP-Na, 10 mM $Na_2$-phosphocreatine, and 2 mM QX314-Cl (295 mosmol, pH 7.35). Membrane potentials were not corrected for liquid junction potential (experimentally measured as 12.5 mV for the $K^+$-based pipette solution and 9.5 mV for the $Cs^+$-based pipette solution).

Neurons were visualized with video-assisted infrared differential interference contrast imaging and fluorescent neurons were identified by epifluorescence imaging under a water immersion objective (40×, 0.8 numerical aperture) on an upright SliceScope Pro 1000 microscope (Scientifica) with an infrared IR-1000 CCD camera (DAGE-MTI). Data were acquired at 10 kHz and low-pass filtered at 4 kHz with an Axon Multiclamp 700B amplifier and an Axon Digidata 1550 or 1440 Data Acquisition System under the control of Clampex 10.7 (Molecular Devices). For the photostimulation of ChR2-expressing neurons, blue light was emitted from a collimated light-emitting diode (LED) of 455 nm. The LED was driven by a LED driver (Mightex) under the control of an Axon Digidata 1550 Data Acquisition System and Clampex 10.7. Light was delivered through the reflected light fluorescence illuminator port and the 40 × objective. Data were analyzed offline using Clampfit 10.7 (Molecular Devices) or AxoGraph X (AxoGraph Scientific).

Neuronal intrinsic excitability was examined with the $K^+$-based pipette solution. The resting membrane potential was recorded in the whole-cell current clamp mode within the first minute after break-in. After balancing the bridge, the input resistance was measured by injecting a 500 ms hyperpolarizing current pulse (10–100 pA) to generate a small membrane potential hyperpolarization (2–10 mV) from the resting membrane potential. Depolarizing currents were increased in 5- or 10-pA steps to identify rheobase currents.

To record unitary connections between inhibitory interneurons and pyramidal neurons, Pv and Sst interneurons were identified by the Cre-dependent expression of tdTomato. Pyramidal neurons were first recorded in whole-cell voltage clamp mode at the reversal potential for excitation (+10 mV) with the $Cs^+$-based patch pipette solution. A nearby Pv or Sst interneuron was subsequently recorded in the whole-cell current clamp mode with the $K^+$-based patch pipette solution. Action potentials were elicited in Pv or Sst interneurons by a train of 6 depolarizing current steps (2 ms, 1–2 nA) at 10 Hz with 15 s intervals between sweeps. Unitary IPSC (uIPSC) amplitudes were measured from the first IPSCs of the average of 30–50 sweeps. We considered a Pv or Sst interneuron to be connected with a pyramidal neuron when the average uIPSC amplitude was at least three times the baseline standard deviation. Spontaneous EPSCs (sEPSCs) in Pv and Sst interneurons were recorded in whole-cell voltage clamp mode at the reversal potential for inhibition (–70 mV) with the $K^+$-based

patch pipette solution. To detect sEPSCs, data were digitally low-pass filtered at 2 kHz offline and events were detected by a scaled-template algorithm (AxoGraph X). The parameters of the template are: length, 5 ms; baseline, 1.5 ms; amplitude, –2 pA; rise time, 0.3 ms; and decay time, 0.7 ms. For voltage clamp experiments, only recordings with series resistance below 20 MΩ were included.

To isolate Pv or Sst interneurons-mediated quantal IPSCs, pyramidal neurons were recorded in whole-cell voltage clamp mode at the reversal potential for excitation (+10 mV) with the $Cs^+$-based patch pipette solution in a modified ACSF containing 4 mM $MgCl_2$ and 0.5 mM $SrCl_2$ without $CaCl_2$. TTX (1 μM), NBQX (10 μM), and CPP (10 μM) were also included in the modified ACSF to block synaptic excitation and reduce overall activity in the slices. Typically, 10–30 sweeps were recorded for each neuron with 40 s intervals between sweeps. During each sweep, mIPSCs were recorded during the 10 s baseline period and the 10 s blue light stimulation period. The light intensity (0.15–7.43 mW/mm$^2$) was ramped down to reduce the tonic currents (*Figure 9—figure supplement 1*). To detect mIPSCs, data were digitally low-pass filtered at 2 kHz offline and events were detected by a scaled-template algorithm (AxoGraph X). The parameters of the template are: length, 20 ms; baseline, 3 ms; amplitude, 2 pA; rise time, 0.6 ms; and decay time, 10 ms. The average amplitude, charge, and decay time constant of quantal IPSCs from ChR2-expressing interneurons were computed as $A_{quantal} = (A_{light}f_{light} - A_{baseline}f_{baseline})/(f_{light} - f_{baseline})$, where $A_{quantal}$, $A_{light}$, and $A_{baseline}$ are the amplitude, charge, or decay time constant of quantal IPSCs, mIPSCs during light stimulation period, and mIPSCs during baseline period, respectively; $f_{light}$ and $f_{baseline}$ are the frequency of mIPSCs during light stimulation period and that of mIPSCs during baseline period, respectively. The average traces of quantal IPSCs were computed similarly using the average traces of mIPSCs from the light stimulation period and baseline period. Only recordings with series resistance below 20 MΩ were included. Data were also excluded if blue light did not significantly evoke more mIPSCs than the baseline period (i.e., p>0.05) or totally less than 150 evoked mIPSCs (i.e., the number of mIPSCs during the blue light stimulation period minus the number of mIPSCs during the baseline period) were obtained. The criterion of 150 evoked mIPSCs was chosen because the likelihood to accurately estimate the parameters (i.e., less than 10% error) from 150 events is higher than 95%.

## Statistics

All reported sample numbers (*n*) represent independent biological replicates that are the numbers of tested mice or recorded neurons. Statistical analyses were performed with Prism 6, 7, or 8 (GraphPad Software). D'Agostino-Pearson, Shapiro-Wilk, and Kolmogorov-Smirnov tests were used to determine if data were normally distributed. If all data within one experiment passed all three normality tests, then the statistical test that assumes a Gaussian distribution was used. Otherwise, the statistical test that assumes a non-Gaussian distribution was used. All statistical tests were two-tailed with an alpha of 0.05. Gender effect was inspected by two-way or three-way ANOVA. The details of all statistical tests, numbers of replicates, and *P* values were reported in *Supplementary file 3*.

## Acknowledgements

This article is dedicated to the memory of Caroline DeLuca, who inspired this project. We thank Gabriele Schuster for the ES cell work and blastocyst injection, Corinne Spencer and James Frost for suggestions and discussions, and Shuyun Deng and Kazuhiro Oka at the Baylor College of Medicine Gene Vector Core for recombinant AAV vector production. This work was supported in part by Citizens United for Research in Epilepsy (CURE Epilepsy Award to MX), the National Institutes of Health (R01NS100893 and R01MH117089 to MX, F30MH118804 to CML), American Epilepsy Society (Postdoctoral Research Fellowship to WC), the Eunice Kennedy Shriver National Institute of Child Health and Human Development (U54HD083092 to Baylor College of Medicine Intellectual and Developmental Disabilities Research Center, Neurobehavioral Core), and the In Vivo Neurophysiology Core of Jan and Dan Duncan Neurological Research Institute. CML and JEM are part of the Baylor College of Medicine Medical Scientist Training Program and McNair MD/PhD Student Scholars supported by the McNair Medical Institute at the Robert and Janice McNair Foundation. HYZ is a Howard Hughes Medical Institute investigator. MX is a Caroline DeLuca Scholar.

## Additional information

### Competing interests
Huda Y Zoghbi: Senior Editor, eLife. The other authors declare that no competing interests exist.

### Funding

| Funder | Grant reference number | Author |
|---|---|---|
| Citizens United for Research in Epilepsy | CURE Epilepsy Award | Mingshan Xue |
| National Institute of Neurological Disorders and Stroke | R01NS100893 | Mingshan Xue |
| National Institute of Mental Health | R01MH117089 | Mingshan Xue |
| Eunice Kennedy Shriver National Institute of Child Health and Human Development | U54HD083092 | Huda Y Zoghbi |
| American Epilepsy Society | Postdoctoral Research Fellowship | Wu Chen |
| Robert and Janice McNair Foundation | McNair MD/PhD Student Scholars | Colleen M Longley Jessica E Messier |
| National Institute of Mental Health | F30MH118804 | Colleen M Longley |

The funders had no role in study design, data collection and interpretation, or the decision to submit the work for publication.

### Author contributions
Wu Chen, Data curation, Software, Formal analysis, Investigation, Visualization, Methodology, Writing - original draft, Writing - review and editing; Zhao-Lin Cai, Formal analysis, Investigation, Visualization, Methodology, Writing - original draft; Eugene S Chao, Data curation, Software, Formal analysis, Investigation, Visualization, Methodology, Writing - original draft; Hongmei Chen, Shuang Hao, Investigation, Writing - review and editing; Colleen M Longley, Formal analysis, Investigation, Visualization, Methodology, Writing - original draft, Writing - review and editing; Hsiao-Tuan Chao, Supervision, Investigation, Writing - review and editing; Joo Hyun Kim, Jessica E Messier, Formal analysis, Investigation, Writing - review and editing; Huda Y Zoghbi, Jianrong Tang, John W Swann, Supervision, Writing - review and editing; Mingshan Xue, Conceptualization, Formal analysis, Supervision, Funding acquisition, Investigation, Visualization, Methodology, Writing - original draft, Project administration, Writing - review and editing

### Author ORCIDs
Wu Chen (iD) http://orcid.org/0000-0002-7400-0519
Zhao-Lin Cai (iD) http://orcid.org/0000-0003-4034-2884
Colleen M Longley (iD) http://orcid.org/0000-0001-8326-6143
Hsiao-Tuan Chao (iD) http://orcid.org/0000-0002-2854-5470
Jessica E Messier (iD) http://orcid.org/0000-0002-5865-7043
Huda Y Zoghbi (iD) http://orcid.org/0000-0002-0700-3349
John W Swann (iD) http://orcid.org/0000-0001-8995-5812
Mingshan Xue (iD) https://orcid.org/0000-0003-1463-8884

### Ethics
Animal experimentation: This study was performed in strict accordance with the recommendations in the Guide for the Care and Use of Laboratory Animals of the National Institutes of Health. All procedures to maintain and use mice were approved in the Animal Research Protocol AN-6544 by the Institutional Animal Care and Use Committee at Baylor College of Medicine.

Decision letter and Author response
Decision letter https://doi.org/10.7554/eLife.48705.sa1
Author response https://doi.org/10.7554/eLife.48705.sa2

## Additional files

### Supplementary files

• Supplementary file 1. EEG phenotypes of individual WT and *Stxbp1*$^{tm1d/+}$ mice. The parameters charactering the SWDs, myoclonic jerks, and myoclonic jumps of each WT and *Stxbp1*$^{tm1d/+}$ mouse are presented in the table.

• Supplementary file 2. Phenotypic comparison of human patients and mouse models. The phenotyping tests in mouse models (the second column) are grouped based on the clinical features of *STXBP1* encephalopathy (the first column). The results of phenotyping tests from different mouse models and studies are presented in the table.

• Supplementary file 3. Statistics of experimental results. The details of all statistical tests, numbers of replicates, and *P* values are presented for each experiment in the table.

• Transparent reporting form

### Data availability

All data generated or analyzed during this study are included in the manuscript and supporting files.

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
