## [Decision Letter]

**Acceptance summary:**

The manuscript by Chen and colleagues characterizes a mouse model of Munc18 haploinsufficiency and presents electrophysiological analysis of inhibitory interneuron function in these mutants that gives rise to the behavioral phenotypes. Deficiencies in synaptic vesicle fusion machinery is being increasingly recognized as key targets for several neurological and neurodevelopmental disorders. The current study provides mechanistic insight into how Munc18, a key component of synaptic vesicle fusion machinery, haploinsufficiency may lead to disease phenotypes.

**Decision letter after peer review:**

Thank you for submitting your article "*Stxbp1* haploinsufficiency impairs cortical inhibition and mediates key neurological features of *STXBP1* encephalopathy" for consideration by *eLife*. Your article has been reviewed by three peer reviewers, one of whom is a member of our Board of Reviewing Editors, and the evaluation has been overseen by Catherine Dulac as the Senior Editor. The following individual involved in review of your submission has agreed to reveal their identity: Markus Missler (Reviewer #3).

The reviewers have discussed the reviews with one another and the Reviewing Editor has drafted this decision to help you prepare a revised submission.

Summary:

In this study, the authors present an extensive behavioral analysis of *STXBP1* (also called munc-18) haploinsufficiency in mice. While this type of analysis has been previously performed by other groups, the current manuscript presents a systematic evaluation of the phenotypes as well as a potential link to alterations in inhibitory neurotransmission in the somatosensory cortex.

Essential revisions:

There are a number of issues that require further attention. Overall, the authors' main claim to novelty is based on the fact that they present more consistent and robust behavioral phenotypes than previous studies that include epileptiform activities, cognitive deficits etc. However, circuit based analysis of synaptic transmission that may uncover the mechanisms underlying these phenotypes is rather cursory. Therefore, this analysis should be expanded as indicated below to address several open questions raised by the findings.

1) The authors should specifically justify and expand the aspect of this work that sets it apart from earlier studies (e.g. Kovacevic et al., 2018; Miyamoto et al., 2019) that address the very same questions.

2) The authors primarily focus on inhibitory neurotransmission by indicating that "…a decrease in excitatory transmission is unlikely adequate to explain how *Stxbp1* haploinsufficiency in vivo leads to cortical hyperexcitability". The logic behind this strong statement is unclear. A decrease in excitatory drive onto inhibitory interneurons, in the least, could contribute to such as phenotype. In this regard, the authors should complement their analysis of GABAergic transmission with analysis of excitatory inputs interneurons expressing Parvalbumin or Somatostatin.

3) The analysis focuses on a reduction in GABAergic neurotransmission originating from cortical inhibitory interneurons expressing Parvalbumin and Somatostatin. While Parvalbumin neurons show a reduction in unitary response amplitudes (no changes in short term plasticity) Somatostatin neurons show a reduction in connectivity. The authors do not present any detailed analysis of these phenotypes. For instance, these effects could easily be postsynaptic and the absence of any in depth analysis complicates a straightforward interpretation. Are there any changes in Sr2+ driven asynchronous events driven by the two inputs? Any differences in spontaneous mIPSCs? These additional parameters will help to strengthen the arguments.

---

## [Author Response]

Essential revisions:There are a number of issues that require further attention. Overall, the authors' main claim to novelty is based on the fact that they present more consistent and robust behavioral phenotypes than previous studies that include epileptiform activities, cognitive deficits etc. However, circuit based analysis of synaptic transmission that may uncover the mechanisms underlying these phenotypes is rather cursory. Therefore, this analysis should be expanded as indicated below to address several open questions raised by the findings.1) The authors should specifically justify and expand the aspect of this work that sets it apart from earlier studies (e.g. Kovacevic et al., 2018; Miyamoto et al., 2019) that address the very same questions.

Previous studies (Miyamoto et al., 2017; Kovačević et al., 2018; Orock et al., 2018) characterized the phenotypes of three lines of *Stxbp1* heterozygous knockout mice. Compared to the current study, previous characterization was limited in scope, used relatively smaller cohorts of mice, and reported some inconsistent results. Here we present a more comprehensive neurological and behavioral study of two new *Stxbp1* haploinsufficiency models and report consistent and robust phenotypes. We feel that these new models are construct and face valid models of *STXBP1* encephalopathy and will be useful for the community to study the disease pathogenesis and explore therapeutic strategies. Furthermore, we report distinct deficits of GABAergic synaptic transmission from two main classes of cortical interneurons. These points are now elaborated in the Introduction and Discussion.

2) The authors primarily focus on inhibitory neurotransmission by indicating that "…a decrease in excitatory transmission is unlikely adequate to explain how Stxbp1 haploinsufficiency in vivo leads to cortical hyperexcitability". The logic behind this strong statement is unclear. A decrease in excitatory drive onto inhibitory interneurons, in the least, could contribute to such as phenotype. In this regard, the authors should complement their analysis of GABAergic transmission with analysis of excitatory inputs interneurons expressing Parvalbumin or Somatostatin.

We agree with the reviewers that a reduction of the excitatory inputs onto inhibitory interneurons could lead to cortical hyperexcitability. Thus, we recorded the spontaneous excitatory postsynaptic currents (sEPSCs) in both Pv and Sst interneurons, but did not observe any significant changes of either amplitude or frequency in the mutant mice, suggesting that the excitatory drive onto interneurons is normal in *Stxbp1* haploinsufficient mice. These results are now presented in Figure 8—figure supplement 2.

3) The analysis focuses on a reduction in GABAergic neurotransmission originating from cortical inhibitory interneurons expressing Parvalbumin and Somatostatin. While Parvalbumin neurons show a reduction in unitary response amplitudes (no changes in short term plasticity) Somatostatin neurons show a reduction in connectivity. The authors do not present any detailed analysis of these phenotypes. For instance, these effects could easily be postsynaptic and the absence of any in depth analysis complicates a straightforward interpretation. Are there any changes in Sr2+ driven asynchronous events driven by the two inputs? Any differences in spontaneous mIPSCs? These additional parameters will help to strengthen the arguments.

We agree with the reviewers that it is crucial to understand if the synaptic transmission deficit of Pv synapses is due to a postsynaptic mechanism. To address this question, we developed a new optogenetic method to isolate quantal IPSCs mediated by the GABA release specifically from Pv or Sst interneurons. We used ChR2 to enhance asynchronous exocytosis of synaptic vesicles from Pv or Sst interneurons in the presence of TTX and Sr^2+^, and then mathematically subtracted the mIPSCs recorded during the baseline period (i.e., before blue light stimulation) from those recorded during blue light stimulation to obtain the average amplitude, charge, and decay time constant of Pv or Sst interneuron-mediated quantal IPSCs. These experiments showed that none of these properties are different between WT and mutant mice, indicating that *Stxbp1* haploinsufficiency does not affect the postsynaptic properties of inhibitory transmission. The results are now presented in Figure 9.

These results indicate that the reduction in the strength of Pv synapses is most likely due to a decrease in the number of readily releasable vesicles or release probability given the role of *Stxbp1* in synaptic vesicle priming and fusion (Rizo and Xu, 2015) and the fact that the quantal amplitude and connectivity are unaltered in *Stxbp1^tm1d/+^* mice. Although the short-term synaptic depression is unaltered in *Stxbp1^tm1d/+^* mice, a change in release probability is still possible because at the Pv interneuron synapses the short-term synaptic plasticity during a short train of action potentials, which was used in this study, is not sensitive to the release probability (Kraushaar and Jonas, 2000; Luthi et al., 2001).